# Optimizing circadian drug infusion schedules towards personalized cancer chronotherapy

**Roger J. W. Hill** [1] *, **Pasquale F. Innominato** [2,3], **Francis Lévi** [3,4], **Annabelle Ballesta** [4]

**1** EPSRC & MRC Centre for Doctoral Training in Mathematics for Real-World Systems, University of Warwick, Coventry, UK, **2** North Wales Cancer Centre, Ysbyty Gwynedd, Betsi Cadwaladr University Health Board, Bangor, UK, **3** Cancer Chronotherapy Team, Cancer Research Centre, Division of Biomedical Sciences, Warwick Medical School, Coventry, UK, **4** INSERM and Paris Sud university, UMRS 935, Team "Cancer Chronotherapy and Postoperative Liver Functions", Campus CNRS, Villejuif, F-94807, France. & Honorary position, University of Warwick, UK

* r.hill.3@warwick.ac.uk

**Data Availability Statement:** All model code is available from a public GitHub repository at https://github.com/Rogerjwhill/Optimizing-circadian-drug-infusion..git.

## Abstract

Precision medicine requires accurate technologies for drug administration and proper systems pharmacology approaches for patient data analysis. Here, plasma pharmacokinetics (PK) data of the OPTILIV trial in which cancer patients received oxaliplatin, 5-fluorouracil and irinotecan via chronomodulated schedules delivered by an infusion pump into the hepatic artery were mathematically investigated. A pump-to-patient model was designed in order to accurately represent the drug solution dynamics from the pump to the patient blood. It was connected to semi-mechanistic PK models to analyse inter-patient variability in PK parameters. Large time delays of up to 1h41 between the actual pump start and the time of drug detection in patient blood was predicted by the model and confirmed by PK data. Sudden delivery spike in the patient artery due to glucose rinse after drug administration accounted for up to 10.7% of the total drug dose. New model-guided delivery profiles were designed to precisely lead to the drug exposure intended by clinicians. Next, the complete mathematical framework achieved a very good fit to individual time-concentration PK profiles and concluded that inter-subject differences in PK parameters was the lowest for irinotecan, intermediate for oxaliplatin and the largest for 5-fluorouracil. Clustering patients according to their PK parameter values revealed patient subgroups for each drug in which inter-patient variability was largely decreased compared to that in the total population. This study provides a complete mathematical framework to optimize drug infusion pumps and inform on inter-patient PK variability, a step towards precise and personalized cancer chronotherapy.

## Author summary

Accuracy and safety of infusion pumps remain a critical issue in the clinics and the development of accurate mathematical models to optimize drug administration though such devices has a key part to play in the advancement of precision medicine. Here, PK data from cancer patient receiving irinotecan, oxaliplatin and 5-fluorouracil into the hepatic

**Funding:** The author RJWH received funding from the grant EP/L015374/1, provided by Engineering and Physical Sciences Research Council, Medical Research Council and the University of Warwick. https://epsrc.ukri.org/ https://mrc.ukri.org/ https://warwick.ac.uk/ The funders had no role in study design, data collection and analysis, decision to publish, or preparation of the manuscript.

**Competing interests:** The authors have declared that no competing interests exist.

artery via an infusion pump was mathematically investigated. A pump-to-patient model was designed and revealed significant inconsistencies between intended drug profiles and actual plasma concentrations. This mathematical model was then used to suggest improved profiles in order to minimise error and optimise delivery. Physiologically-based PK models of the three drugs were then linked to the pump-to-patient model. The whole framework achieved a very good fit to data and allowed quantifying inter-patient variability in PK parameters and linking them to potential clinical biomarkers via patient clustering. The developed methodology improves our understanding of patient-specific drug pharmacokinetics towards personalized drug administration.

## Introduction

Cancer management is challenged by large inter- and intra-patient variabilities in both disease progression and response to treatments. Thus, the quest for accurate and personalized cancer therapies has fostered the development of new technologies enabling multi-type measurements in individual patients and complex drug scheduling. To translate datasets available for an individual patient into personalized therapies and further ensure their precise administration, new mathematical approaches are required. Indeed, systems medicine, that involves the implementation of theoretical approaches in medical research and practice, is critically needed as emphasized in the roadmaps of the Coordinated Action for Systems Medicine (CaSyM) from the European Union (https://www.casym.eu, [1]) and of the Avicenna action (http://avicenna-isct.org/), and in other international consortia [2–5]. The final aim is a measurable improvement of patient health through systems-based practice which will enable predictive, personalised, participatory and preventive (P4) medicine [6].

Accuracy and safety of infusion pumps are mandatory to ensure that the correct drug dose is delivered to the patient over the intended period. Recurrent incidents related to devices delivering fluids such as nutrients or medications into the body have led the U.S Food and Drug Administration (FDA) to launch in 2010 an initiative to reduce infusion pump risks (https://www.fda.gov/medicaldevices/productsandmedicalprocedures/generalhospitaldevicesandsupplies/infusionpumps/ucm202501.htm). Many of the reported events are related to deficiencies in the initial design of the device and of the embedded software. Adverse events may also arise from a defect appearing over the device's life cycle due to technical failure or lack of proper maintenance. However, due to the complexity of the equipment, user errors are also common [7].

Optimizing chemotherapeutics index, defined as the ratio between treatment antitumor efficacy and induced toxicities, is complex at multiple levels. First, large inter-patient variabilities are demonstrated in drug pharmacokinetics, tolerability and anti-tumour efficacy [2, 8–10]. Next, important intra-patient variabilities arise from the fact that tumour and healthy tissues, rather than being static over time, display time-dependent variations, in particular over the 24h span, which are called circadian rhythms [11]. The circadian timing system controls most physiological functions of the organism resulting in drug Absorption, Distribution, Metabolism and Elimination (ADME) displaying 24h-rhythms with differences of up to several folds between minimum and maximum activities [12, 13].

Chronotherapy -that is administering drugs according to the patient's biological rhythms over 24 h- is a growing field in medicine and especially in oncology. Indeed, at least 22 clinical trials involving a total of 1773 patients with different types of metastatic cancers have demonstrated a significant influence of administration timing on the tolerability of 11 commonly-used

antitumor drugs [14]. Two randomized phase III clinical trials in 278 metastatic colorectal cancer (mCRC) patients receiving oxaliplatin and 5-fluorouracil showed that cancer chronotherapy achieved an up-to-5-fold decrease in treatment side effects and nearly doubled anti-tumour efficacy compared to conventional administration of the same drug doses [15]. However, a meta-analysis of these two studies combined to another clinical trial involving 564 mCRC patients receiving the same drugs (497 men and 345 women in total) concluded that the chronomodulated drug modality significantly increased the efficacy and survival in men while reducing that in women as compared to conventional administration [16]. Such sex-specificity was further validated for irinotecan chronotoxicity in mouse experiments [17] and in a clinical trial involving 199 mCRC patients treated with oxaliplatin (infusion peak 4pm), 5-fluorouracil (infusion peak 4am) and irinotecan given at 6 different circadian times [18]. Both studies showed a higher circadian amplitude in females as compared to males and a difference of several hours between the optimal timing of each gender. Furthermore, circadian biomarker monitoring in individual patients recently revealed up to 12 h inter-patient differences regarding the timing of midsleep, the circadian maximum in skin surface temperature or that in physical activity [19]. These investigations have highlighted the need for the individualization of drug combinations and chronoinfusion schemes to further improve treatment outcome, taking into account the patients' sex, chronotype and genetic background. The accurate delivery of intended administration profiles is obviously critical in this context. Chronotherapy requires the error in drug infusion timing not to be greater than few hours.

Clinical findings about cancer chronotherapy have motivated the development of innovative technologies for chronomodulated drug delivery including the Mélodie infusion pump (Axoncable, Montmirail, France, [20]). This portable electronic pump allows for the administration of up to 4 compounds according to pre-programmed schedules over the 24 h span. It was used in several clinical trials for the chronomodulated delivery of irinotecan (CPT11), oxaliplatin (L-OHP) and 5-fluorouracil (5-FU) into the central vein of metastatic colorectal cancer patients [13]. The Mélodie pump was recently used to infuse those three anticancer drugs directly into the hepatic artery of metastatic cancer patients in the translational European OPTILIV Study [20]. This uncommon delivery route into the hepatic artery and the use of an infusion pump to deliver the drugs according to chronomodulated profiles represent a novel chemotherapeutic approach which needs to be quantitatively investigated to maximize patient benefit. In this study, the plasma pharmacokinetics of oxaliplatin revealed inconsistencies between programmed delivery schedules and observed drug concentration within the patient blood including a delay in the time taken for the drug to be detectable in the blood and unexpected peaks in plasma concentrations during drug infusion. Such inconsistencies between targeted drug exposure patterns and plasma drug levels motivated the design of a mathematical model of fluid dynamics within the pump system presented hereafter. This pump-to-patient model was then connected to semi-physiological PK models to investigate the inter-patient variability in drug PK after hepatic artery administration. Thus, this systems pharmacology study aimed to develop predictive mathematical models allowing for the quantitative and general understanding of i) the pump dynamics, irrespective of the drug delivery device, and ii) patient-specific whole-body PK of irinotecan, oxaliplatin and 5-fluorouracil after drug administration using an infusion pump. Such mathematical techniques would then allow for precise and personalized drug timing.

## Results

The overall objective of this study was to accurately investigate the inter-patient variability in the plasma PK of the three anticancer drugs administered during the OPTILIV trial. A first

strategy consisted in using compartmental PK modelling taking the delivery profiles programmed into the infusion pump as inputs for the plasma compartments. However, such methodology revealed inconsistencies between the best-fit models and the data, including delays of several hours. We then concluded that the fluid dynamics from the pump to the patient had to be quantitatively modelled. Hence, we designed the complete model in two sequential mathematical studies. First, we studied the drug solution dynamics from the pump to the patient blood for which the model was based on partial differential equations. This novel model of the pump delivery system took into account the specificity of the equipment used in order to accurately predict drug delivery in the patients' blood, although it those can be easily adapted to any drug delivery devices. Second, we connected this model to compartmental PK models based on ordinary differential equations. This complete framework allowed for the investigation of inter-patient variability in drug PK after hepatic artery administration.

## Pump-to-patient drug solution dynamics

**Model design.** The pump-to-patient model is a transport equation representing the dynamics of the drug solution along the administration tube, with respect to time ($t$) and one-dimensional space ($x$)(Eq 1). $x$ is the distance along the tube from the pump ($x = 0$) to the patient ($x = L$). The drug solution was assumed to be incompressible so that the fluid velocity was considered as constant along the whole tube. Thus, the drug concentration in the tube $u(x, t)$ changes with respect to the following equation:

$$\frac{\partial u(x, t)}{\partial t} = -V(t) \frac{\partial u(x, t)}{\partial x} \qquad t \in [0, T], \quad x \in [0, L] \tag{1}$$

with a Dirichlet boundary condition of,

$$u(0, t) = \frac{S(t)}{sa \times V(t)}, \tag{2}$$

where $V(t)$ is the fluid velocity inside the tube, expressed in m/h. The constant $sa = \pi r^2$ is the cross sectional surface area of the tube (in m$^2$), with $r$ being the radius of the tube. The source term $S(t)$ represents the amount of drug delivered according to the infusion profile programmed into the pump and is expressed in mol/h. Initial conditions along the tube are $u(x, 0) = [0, L]$. The fluid velocity and source terms are controlled by the pump which imposes a fluid delivery rate expressed in ml/h. They are computed by converting the fluid delivery rate into m/h and mol/h respectively using the tube geometry and the concentration of each drug solution. Hence, model simulations at the end of the tube ($x = L$) do not depend on the exact geometry of the tube but rather on its total volume. The input function for PK models depending only on quantities at the end of the tube, the original infusion tube which was constituted of two sections of different diameters was simplified in numerical simulations to a tube of radius 1mm and total length 2340mm that had the same total volume as the original set-up. The total tube volume was set to 1.84 mL as in the equipment used in the OPTILIV study. The transport equation with associated initial and boundary conditions can be solved using the classical method of characteristics which gives [21]:

$$u(t, x) = \begin{cases} 0 & \text{if} \quad \int_0^t V(r) dr < x \\ \frac{S(\tau(t,x))}{sa \times V(\tau(t,x))} & \text{otherwise} \end{cases},$$

where $\tau(x, t)$ is the time at which the drug reaching point x at time $t$ initially entered the system

i.e.

$$\int_{\tau(t,x)}^{t} V(s)ds = x.$$

The input function for the PK models corresponds to the rate of drug infusion into the patient (i.e. at $x = L$) and can be obtained by:

$$d(t) = sa \times V(t)u(t,L) = \begin{cases} 0 & \text{for} \quad t \quad \text{such that} \quad \int_0^t V(r)dr < L \\ V(t)\frac{S(\tau_L(t,L))}{V(\tau_L(t,L))} & \text{otherwise, with} \quad \int_{\tau(t,L)}^t V(s)ds = L \end{cases}$$

Note that, for all drug infusion apart from the glucose flushes, the source term S(t) is proportional to the fluid velocity $V(t)$ as the drug is infused within the tube in the same time as the fluid, so that $d(t)$ is proportional to $V(t)$ once the tube is filled i.e. for times t such that $\int_0^t V(r)dr < L$. An example of the PDE model simulations in time and space for oxaliplatin delivery is shown in Fig 1A.

## Differences between programmed infusion profiles and actual drug delivery in the patient's blood

The pump infusion schemes used in the OPTILIV trial were simulated for the three drugs: irinotecan, oxaliplatin and 5-fluorouracil. Whereas the drug profiles programmed into the pump followed a smooth sinusoidal function, the actual drug delivery in the patient artery differed from the programmed profiles by two main features. First, the model predicted a significant time delay between the actual start of the drug delivery by the pump and the time the drug first reached the patient blood (Fig 1B–1G). This delay was evaluated by the model to 3 h 05 min for oxaliplatin, 2 h 20 min for 5-fluorouracil and 51 min for irinotecan. It corresponded to the time taken to fill the infusion tube with the solution containing the drug at the beginning of the infusion. The delay was drug-specific as it depended on the drug solution concentration and the velocity of the solution in the tube driven by the programmed input profiles. Next, at the end of the infusion profiles, the pump stopped and did not administer the amount of drug left inside the tube. This remaining drug was flushed out by the glucose rinse subsequent to drug administration which induced a sudden delivery spike in the patient artery (Fig 1B–1G). The amount of drug in this spike was expressed in percentage of total drug delivered and was estimated to 10.7% for oxaliplatin, 5.36% for 5-fluorouracil and 1.85% for irinotecan. Doses and rates can be seen in Table 1.

Our systems approach revealed important differences between the intended drug infusion profile and the actual administration into the patient artery. Hence, we developed optimized infusion profiles that strictly achieved the drug administration intended by clinicians. The same equipment was considered to avoid cost of changing. Drug concentrations of the infusion solutions were kept unchanged in order to avoid possible problems of drug stability. In order to administer the drug in the patient's blood following a smooth sinusoidal function, a profile in three parts is required as follows (Fig 2). The first part of the profile is an initial bolus to fill the tube between the pump and the patient with the drug solution. Once the tube is filled, the original sinusoidal profile starts. Then, to solve the problem of the amount of drug left in the tube when the pump stops, the original sinusoidal profile needs to be interrupted when the total drug amount has left the drug bag. Then, a subsequent glucose rinse needs to be infused according to the final segment of the sinusoidal curve in order to deliver the drug remaining in the tube at the correct rate.

**Fig 1.** (A) shows oxaliplatin concentration profile in the infusion tube. The x-axis represents the distance along the tube, the y-axis represents the time from the start of the pump delivery. For figures (B-G), the x-axis represents Clock time and starts at the beginning of the considered drug administration. The left column shows the difference between the intended delivery profiles and the simulated delivery profiles evaluated at the end of the tube (x = L), for irinotecan (B), oxaliplatin (D) and 5-fluorouracil (F). The right-hand column shows the cumulative percentage of drug delivered to the patient for the intended and actual profiles over time for irinotecan (C), oxaliplatin (E) and 5-fluorouracil (G).

**Inter-patient variabilities in irinotecan, 5-fluorouracil and oxaliplatin PK after chrono-modulated administration.** The pump-to-patient model provided educated predictions of the drug infusion into the patients' blood, which was a prerequisite to study the inter-patient variability in the PK of irinotecan, oxaliplatin and 5-fluorouracil. A compartmental

**Table 1. Table describing the defining delivery values for CPT11, LOHP and 5-FU.** The main peak refers to the maximum flow rate from the intended delivery schedule. The spike peak rate refers to the maximum flow rate of the delivery caused by the glucose flush.

| Drug | Total dose (mg/m$^2$) | Drug solution concentration (mg/ml) | Main peak rate (ml/m$^2$/h) | Spike peak rate (ml/m$^2$/h) |
|---|---|---|---|---|
| CPT11 | 180 | 3.33 | 18.02 | 7.38 |
| LOHP | 28 | 3 | 1.63 | 7.28 |
| 5-FU | 933 | 50 | 3.4 | 6.96 |

physiological model was designed for each drug separately, since interactions between CPT11 and LOHP, and between LOHP and 5-FU have not been demonstrated [22, 23] and CPT11 and 5-FU also showed no interaction if CPT11 is delivered first as it is in this study [24]. All parameters were fitted for each patient and each drug independently.

## Compartmental models of irinotecan, oxaliplatin and 5-fluorouracil pharmacokinetics

**Model design.** PK models represented the drug fate in: the Liver, to accurately represent hepatic delivery, the Blood, the measurement site, and the rest of the body known throughout this paper as Organs. The volume of each compartment was individualised for each patient using Vauthey method for Liver [25], Nadler's formula for Blood [26], and Sendroy method for

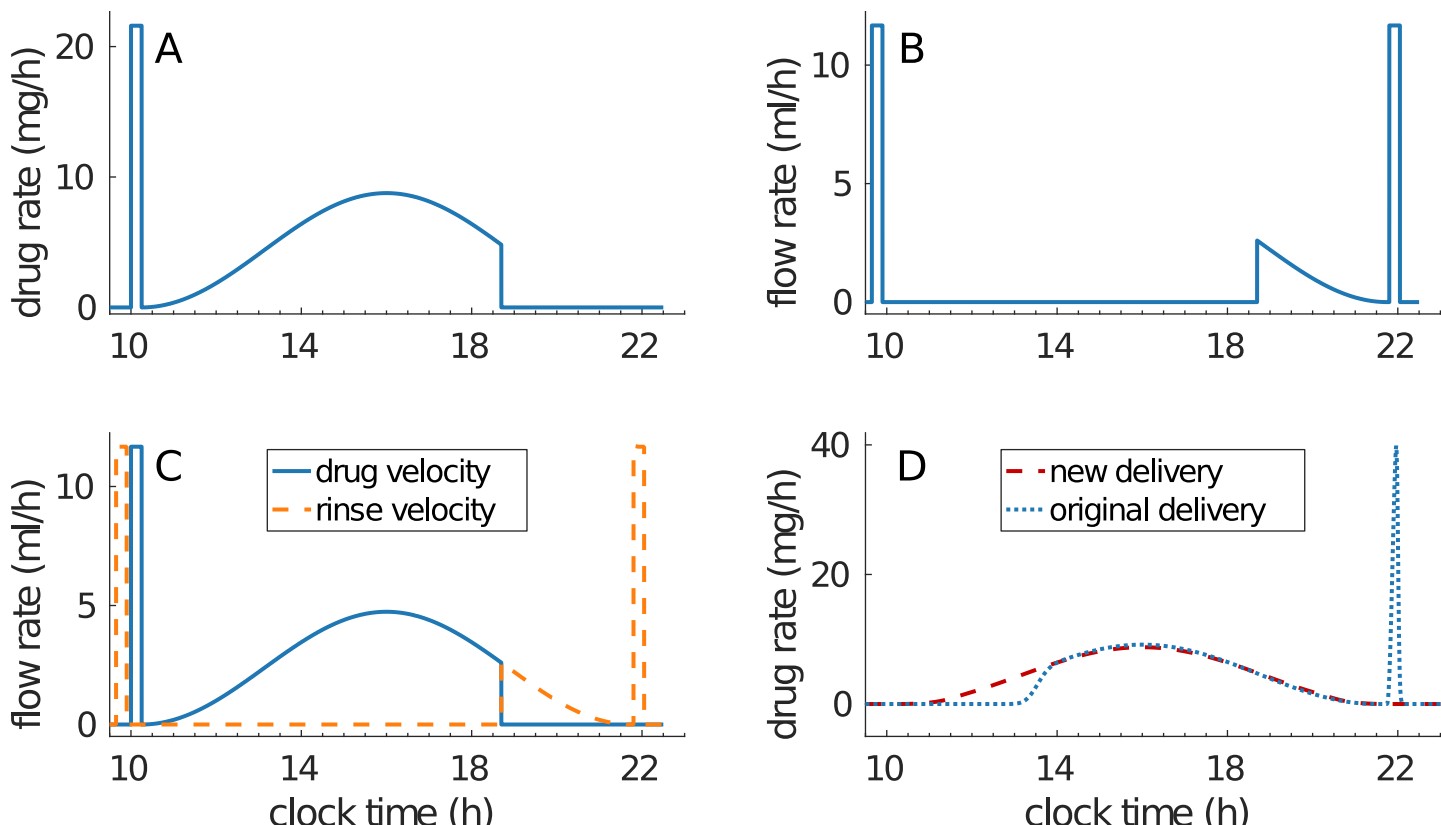

**Fig 2. Improved administration profiles.** (A) shows the drug solution delivery profile which consists of an initial bolus to fill the tube entirely, followed by the original profile. (B) shows the rinse solution delivery rate which continues drug delivery at correct rate while clearing the tube from any active substance, Original rinse peaks were kept unchanged, although there are not mandatory in this administration design. (C) shows how the flow rate along the tube is smoothly switched between the drug and the rinse and (D) shows the new drug delivery profile that will enter the patient compared to the original profile used in the OPTILIV study.

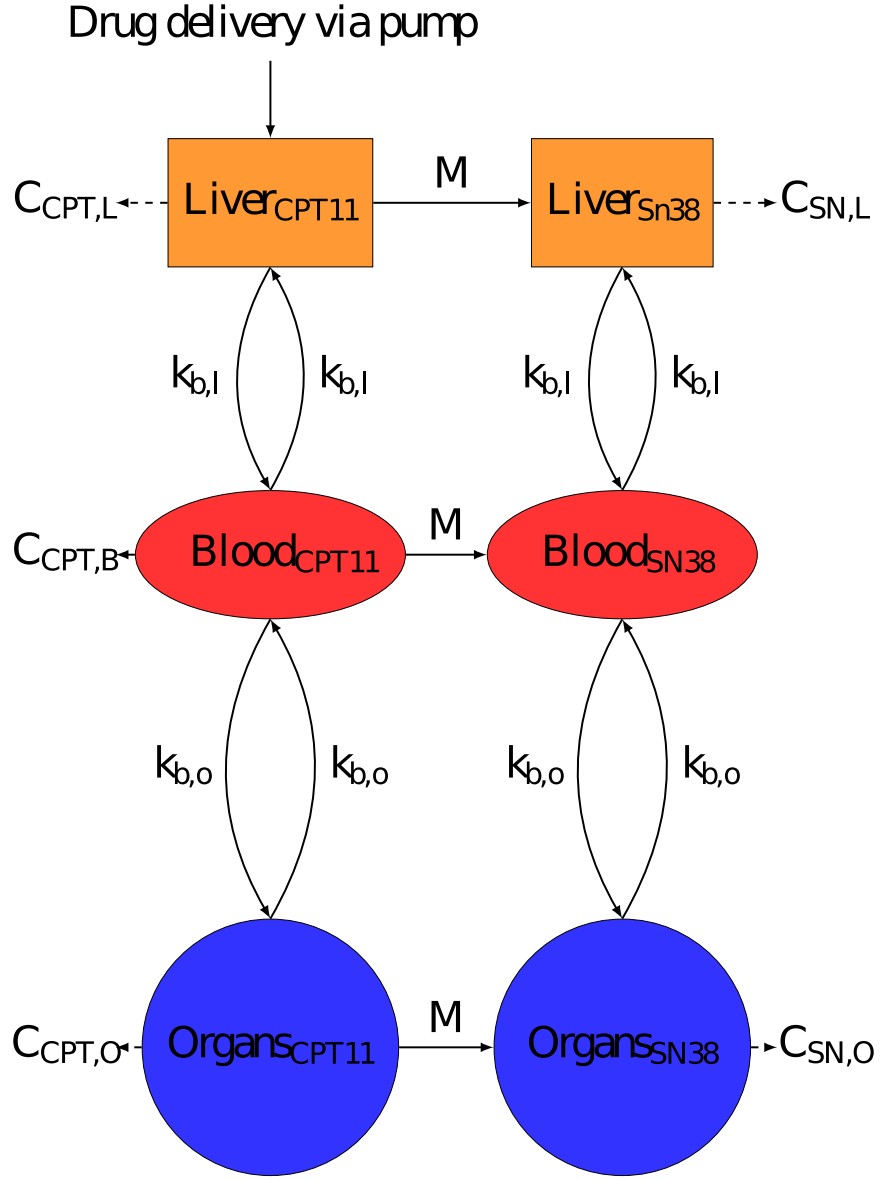

**Fig 3. Semi-physiological model of irinotecan PK.** Compartments were minimised to the most important components, Liver to accurately represent drug delivery, Blood which is measurement site and Organs to represent the rest of the body. $C_i$ is the rate constant of clearance from compartment i. Irinotecan is bio-activated into its active metabolite SN38. Irinotecan was assumed to be delivered directly into the liver.

Organs [27] (See S1 Text for exact formulas used). Each model assumed that the drug was delivered directly into the liver compartment to represent the Hepatic Artery Infusion (HAI, Figs 3, 4 & 5). All transports in between compartments were considered as passive and were represented by linear terms. Throughout this paper, the terms Blood-Liver or Blood-Organ transport represent a bidirectional transport that encompasses the transfer across the blood vessel walls into/from tissues. This simplification has been adopted due to lack of data on the transport processes between compartments. In the models, the drug clearance terms accounted for all types of drug metabolism which were not explicitly modelled (e.g. hepatic CYP450 activity) and i) renal elimination for the Blood compartment, ii) intestinal elimination for the Organs

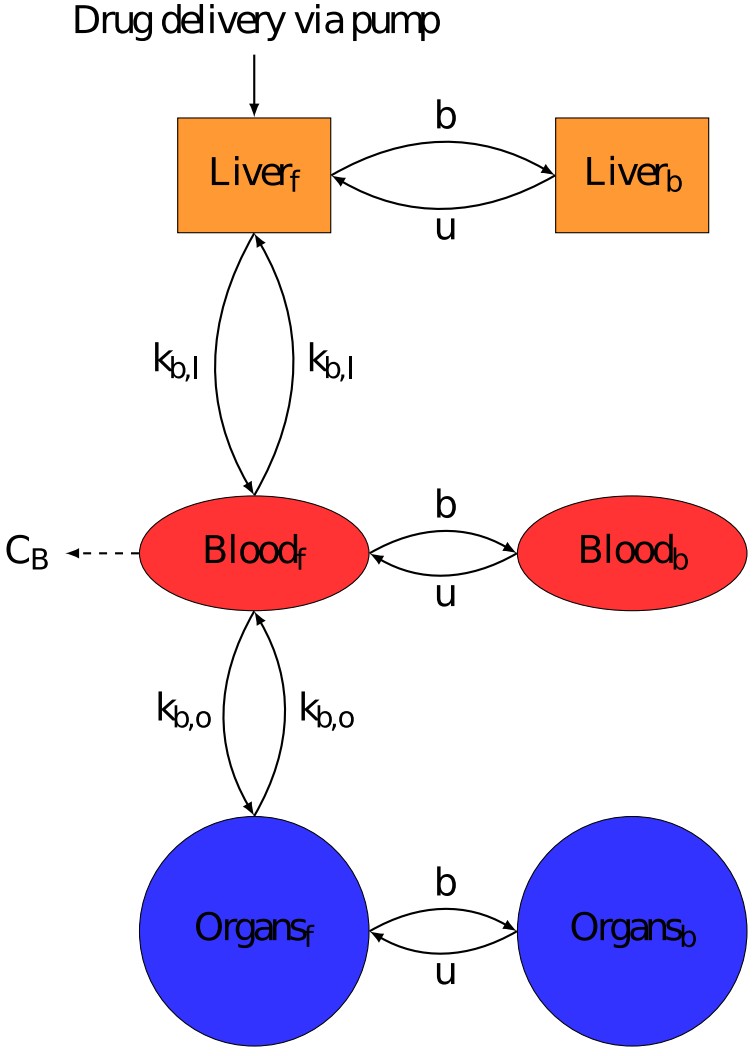

**Fig 4. Semi-physiological model of oxaliplatin PK.** Compartments were minimised to the most important components, Liver to accurately represent drug delivery, Blood which is measurement site and Organs to represent the rest of the body. $C_i$ is the rate constant of clearance from compartment $i$. Each compartment contains a bound and unbound drug fraction and only unbound molecules can migrate between compartments. b and u are respectively the binding and unbinding rate constants of platinum to proteins. Oxaliplatin was assumed to be delivered directly into the liver in its unbound form.

compartment and iii) biliary excretion for the Liver compartment only for irinotecan and 5-fluorouracil since it could be neglected for oxaliplatin [28–30]. The Organs compartment did not include the intestinal lumen and only accounted for the intestinal cells composing the wall of the intestine which were exposed to the drug through blood circulation. The intestinal cells may expel the drug toward the lumen or transform the drug through metabolism, both phenomena being represented by the intestinal clearance in the models. The drug excreted through the bile directly reached the intestinal lumen—which was not considered as part of the Organs compartment- and the drug recirculation was neglected. In the absence of quantitative data and to avoid model over-parametrization, circadian rhythms were neglected in the PK models and all parameters were assumed to be constant over the 8-hour time window of PK measurements.

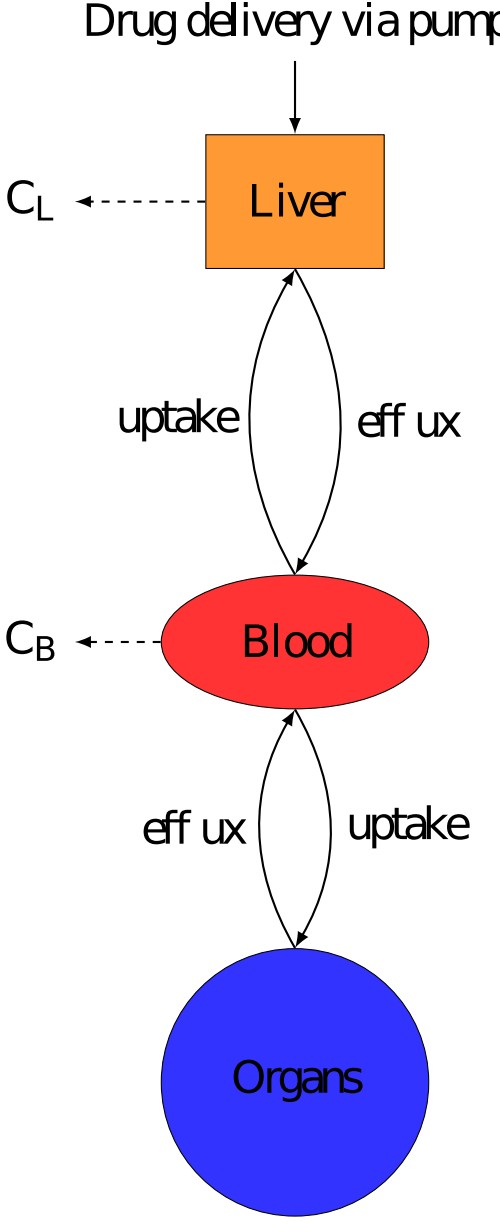

**Fig 5. Semi-physiological model of 5-fluorouracil PK.** Compartments were minimised to the most important components, Liver to accurately represent drug delivery, Blood which is measurement site and Organs to represent the rest of the body. $C_i$ is the rate constant of clearance from compartment *i*. 5-fluorouracil was assumed to be delivered directly into the liver.

Any chemical species bound either to plasma proteins or to DNA was assumed to be unable to move between compartments or to be cleared from the system.

Parameter identifiability assessed though sensitivity analysis to cost function variations revealed poor sensitivity of the clearance rate constant in the Organs compartment for the three drugs (cf. Methods). Hence, Organ clearance was neglected for 5-fluorouracil which is mainly cleared through hepatic metabolism, biliary excretion and renal elimination [30]. Organs clearance and liver clearance was neglected for oxaliplatin since majority of platinum is cleared via renal clearance and the total amount cleared after the end of treatment was set to

54% in line with the literature [28]. Irinotecan organ clearance was assumed to be scaled relative to that of the Liver compartment, this is since similar amounts of irinotecan are cleared via faecal clearance and biliary clearance [29]. In the model of 5-fluorouracil, poor sensitivity was also obtained for transport parameters between Blood and Organs. Hence, transport rate constants were assumed to be proportional to compartment volumes for Blood-Liver and Blood-Organs transport, for each of the three drugs, thus neglecting organ-specific transporter expression.

Parameter likelihood profiles analysis revealed that additional constraints were needed to ensure the identifiability of all parameters (see Methods). Hence, information on renal, intestinal and hepatic clearance relative rates was inferred from literature as follows. For irinotecan, CPT11 drug amount though renal clearance and though combined intestinal elimination and biliary clearance were respectively set to 25% and 60% of the total administered dose [29]. As SN38, which is the active metabolite of CPT11, renal elimination was documented as negligible, the metabolite was considered to only be cleared through Liver, via metabolism into SN38G, or Organs and these cleared amounts were assumed to account for 15% of the total administered dose of irinotecan [29]. The amount of SN38 cleared via metabolism in the liver accounted for approximately 4% of the total administered does of irinotecan whereas SN38 excretion into the intestinal lumen accounted for approximately 9% of the total dose of irinotecan. Therefore we have set the SN38 clearance via Organs to be twice that of the liver clearance [29]. Oxaliplatin clearance was set such that 54% of the total administered drug amount was cleared via the kidneys [28]. The amount of platinum (Pt) bound within the Organs or within the Liver was set to 84% and 12% of the total dose, respectively [31]. The Boughattas et al paper was used to give tissue concentrations, no data to our knowledge exists for humans so we have used this mouse data as a best approximation. The amount of platinum in the tissues was calculated from total amount found in the respective organs relative to total dose. 5-FU was shown to be mainly cleared through hepatic metabolism, so that the amount of drug cleared through the Liver was assumed to account for approximately 80% of the total dose [30].

The final irinotecan model had six compartments as each of the three Liver, Blood and Organs, had two sub-compartments: the parent drug irinotecan, and its active metabolite SN38 (Fig 3). Initial irinotecan administered in the liver was assumed to be only in the form of the parent drug. Irinotecan was converted into SN38 via Michaelis Menten kinetics within the liver and organs, but not in the Blood since the activation enzymes carboxylesterases are not expressed in blood cells in humans [32]. The parameter estimate $K_m = 59.2\mu M$ which reflects the affinity of the substrate and the enzyme was taken directly from an in vitro study in human liver cells [33], thus making the assumption that $K_m$ values are unchanged from in vitro to in vivo as classically done in the literature [34, 35]. SN38 was considered to only be present in its bound form since the bound fraction is reported to be greater than 95% [36]. SN38 clearance terms accounted for SN38 elimination including its deactivation into SN38G though UDP-glycosyltransferases (UGTs) [37].

The oxaliplatin PK model had six compartments corresponding to bound and free (Pt) molecules in the Liver, Blood and other Organs. Oxaliplatin is rapidly metabolised into platinum complex forms [28], which were not distinguished in the current data. In the absence of any data on the dynamics of these different metabolites, they were all assumed to have the same PK properties in the model. Initial oxaliplatin administered in the liver was assumed to be free. Free Pt could bind to proteins and unbind from proteins, due to protein degradation [28], which was included in all compartments (Fig 4).

The final model for 5-fluorouracil had three compartments. The drug clearance accounted for both drug elimination and drug metabolism in each compartment (Fig 5). Protein binding

of 5-fluorouracil was neglected in the model because of the low protein affinity of this drug [38]. Equations for the three models can be seen in S1 Text.

**Inter-patient variability in irinotecan, oxaliplatin and 5-fluorouracil PK parameters.** Overall, each of the three drug models showed a very good fit to data as demonstrated by $R^2$ values averaged over all patients of 0.98 for irinotecan, 0.96 for oxaliplatin and 0.8 for 5-fluorouracil (Figures A, B and C and table D, G and J in S1 Text). These results obtained using infusion rates computed through the pump-to-patient model were compared with simulations with infusion rates equal to the profiles programmed into the pump. Using the pump-to-patient model allowed the improvement of the model fit to pooled data for each drug (Tables C, D, F, G, I and J S1 Text) and the model fit to patient specific data for SSR values by an average of 4.9% for irinotecan, 43.4% for oxaliplatin and 12.5% for 5-fluorouracil, thus proving the validity of our approach. The irinotecan model had an almost perfect fit, it matched the linear increase of AUC compared to dose as described in the FDA drug label (Figure B in S1 Text) [39], and showed a rapid accumulation of both irinotecan and SN38 in the plasma of patients (Fig 6). No obvious impact on irinotecan and SN38 plasma concentrations was observed regarding the time needed to fill the infusion tube or the 30-min glucose delivery spike, as predicted by the pump-to-patient model.

The fit for the oxaliplatin PK model captured all general trends (Fig 7). The model fit for patient 7 did not fully captured the dynamics of total Pt plasma concentration but correctly simulated free Pt concentration. The model did predict i) a delay in plasma Pt concentrations at the start of the infusion due to the pump-to-patient drug transport and ii) a spike during the glucose flush for all patients. This drug spike had an effect on the time of maximum concentration tmax of the free Pt by shifting the time by up to 6 h. The model underestimated the free platinum peak concentrations after the glucose flush for the patients with the most significant rise in concentration, that are patients 2, 3 and 7.

The 5-fluorouracil model showed a very good fit to data, despite a slight systematic underestimation of the third data point in time. It predicted the glucose flush to induce a late spike in plasma drug concentration which could not be seen in the data for all patients, probably because blood sampling frequency was not high enough (Fig 8). This model-predicted spike in 5-fluorouracil concentration changed the tmax value for Patient 5, 6 and 9. The predicted spike AUC was equal to approximately 5% of the total AUC which was in agreement with the pump-to-patient model prediction. This was only calculable for 5-fluorouracil since its elimination was fast enough for its concentration to be close to zero by the time the glucose flush began.

The model fit to each individual patient PK data allowed to investigate the inter-patient variability in resulting PK parameters (Fig 9A, 9B and 9C). The CV of each PK parameter was calculated among the patient population (Table A, E and H in S1 Text). Then, the mean CVs for the entire parameter set of each drug model were calculated as a single measure of inter-patient variability. Irinotecan had the smallest mean CV with a value of 79.18%, and a range from 42.48 to 176.25%. Oxaliplatin had the second smallest value of mean CV, 97.56%, with the largest range from 38.1–318.2%. 5-fluorouracil had the largest mean CV at 112.10%, with the smallest range from 59.4 to 187.51%. In all three models the parameters which showed the largest inter-patient variability were transport parameters specifically, for irinotecan Blood-Organ transport, for oxaliplatin Blood-Liver Transport and Blood-Liver/Organ transport for 5-fluorouracil.

For each drug model, individual patient parameter sets were then utilized to identify patient clusters. The numbers of clusters were determined by minimising the validity index of Fuku-yama and Sugeno $V_{FS}$ as described in [40]. Clustering for different numbers of clusters and their respective $V_{FS}$ can be seen in the S1 Text (Figures G, H and I in S1 Text). For irinotecan,

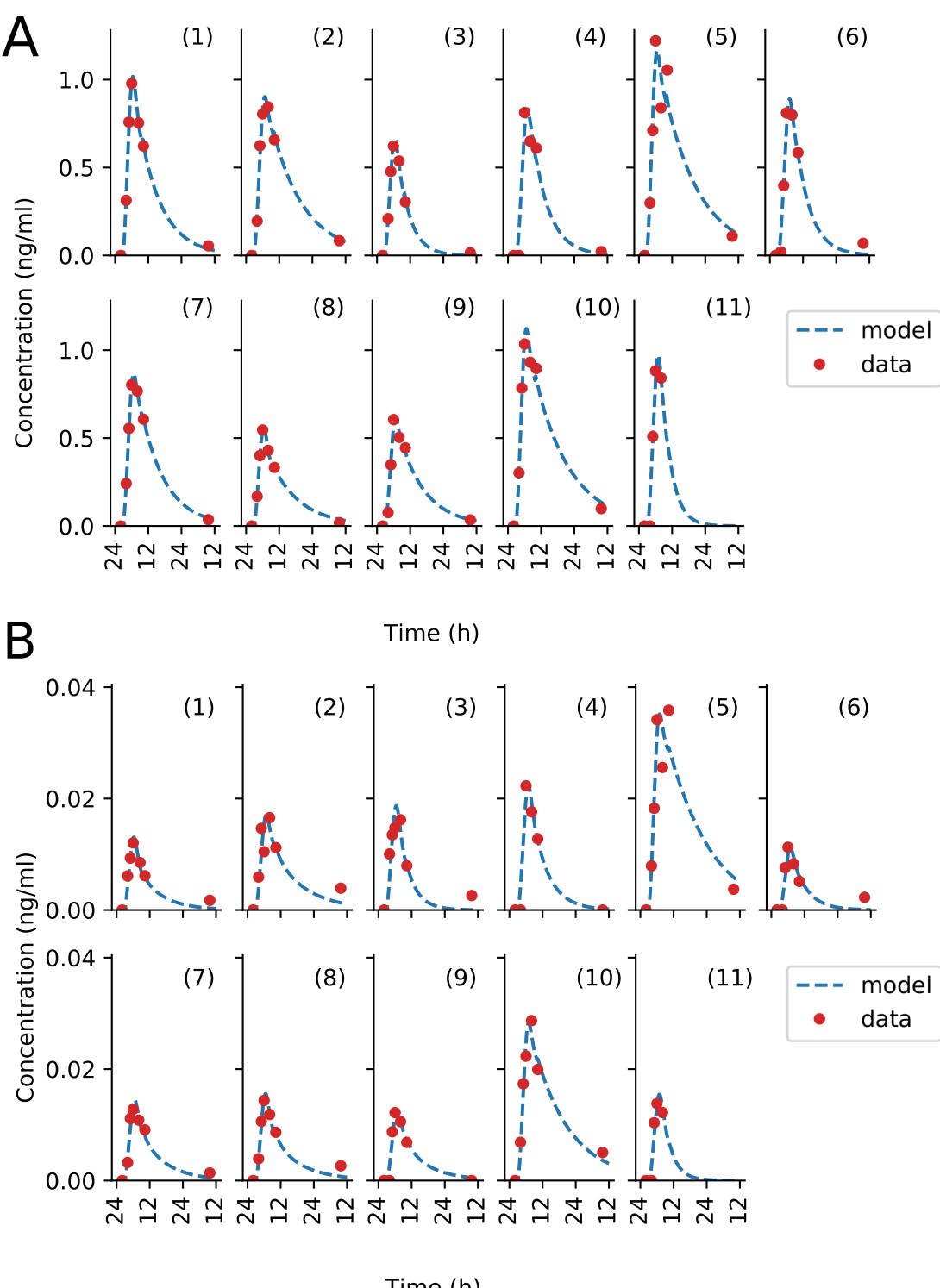

**Fig 6. Patient data best-fit of irinotecan PK model.** Each subplot represents an individual patient dataset, fit to the model independently. (A) shows the fit of irinotecan plasma concentration, (B) shows that of SN38, the active metabolite of irinotecan.

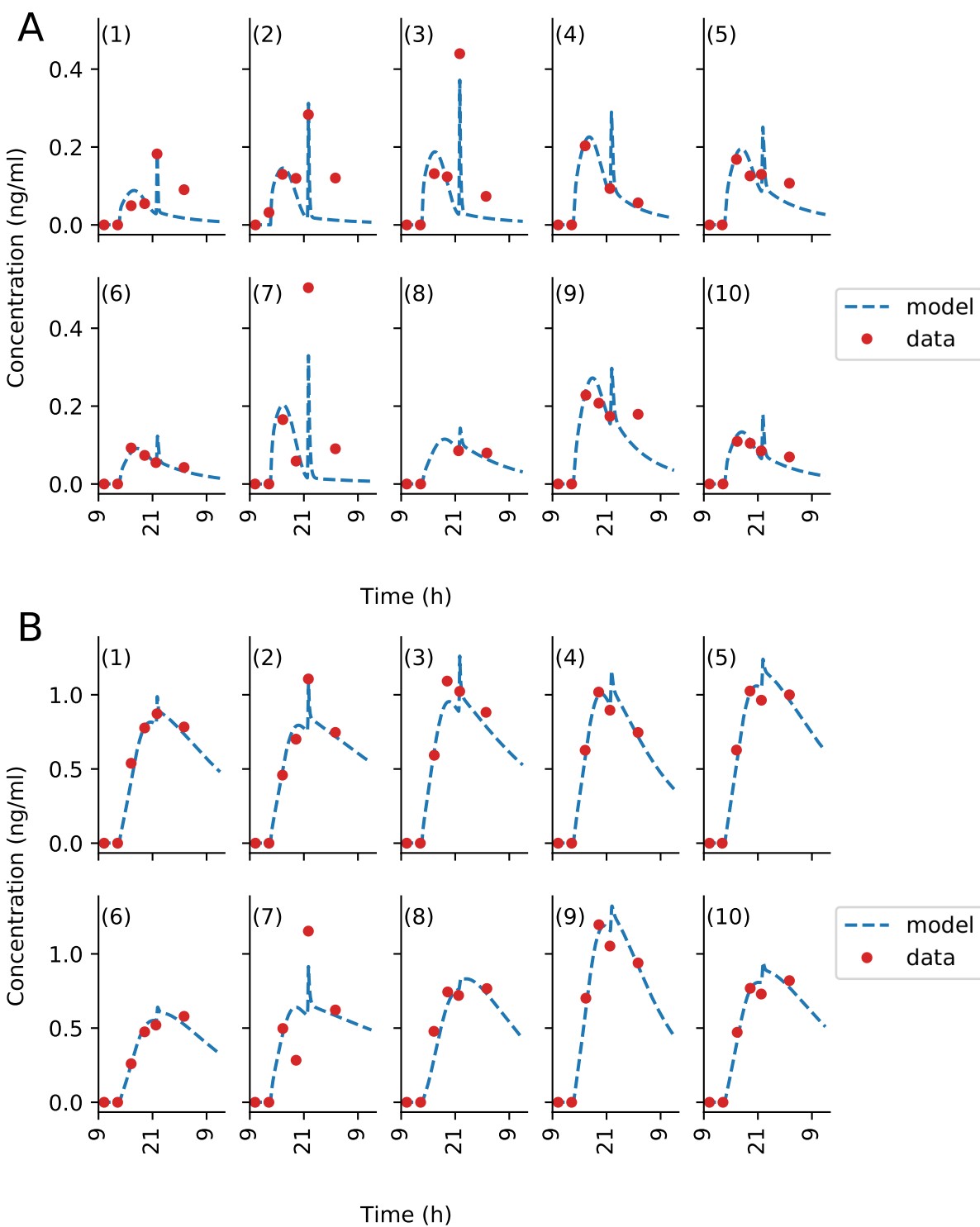

**Fig 7. Patient data best-fit of oxaliplatin PK model.** Each subplot is an individual patient data, fit to the model independently. (A) shows plasma ultrafiltrate platinum concentrations, and (B) shows plasma total platinum concentrations. PK data for Patient 11 was missing.

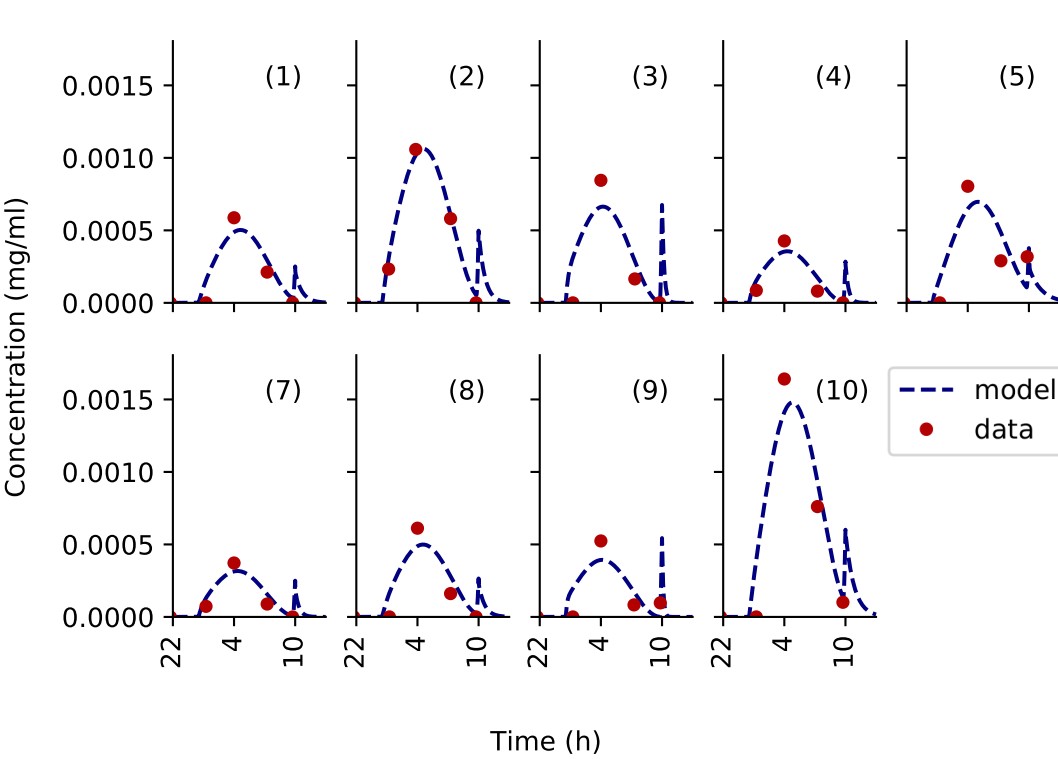

**Fig 8. Patient data best-fit of 5-fluorouracil PK model.** Each subplot is an individual patient data fit to the model independently. PK data for Patient 6 and 11 was missing.

the minimum value of $V_{FS}$ was achieved for four clusters. One cluster was composed of Patients 1, 2, 3, 5, 7, 8, 9 and 10, the other three patients were in a cluster on their own. The analysis for oxaliplatin concluded to two clusters, a cluster of only one patient, patient 7, and the rest of the patients being clustered together. The analysis for 5-fluorouracil revealed four clusters: 5 patients were grouped in the largest cluster (Patients 1, 2, 3, 7, and 10), two patients in the second cluster (Patients 4, 5) and the final two patients were in clusters on their own. Only patients 1, 3 and 10 were consistently clustered together for all three drugs. Once the patient PK parameters had been clustered, the mean of parameter CVs was reassessed for each cluster with 2 or more patients within. Irinotecan mean CV in the largest cluster was 51.52%, which represented a large decrease compared to the mean CV in the entire patient population equal to 79.18%. Oxaliplatin main cluster which was constituted of all patient but patient 7 had a mean CV of 87.37% as compared to 97.56% for the entire population. 5-fluorouracil's largest cluster had a CV of 32.37% and the smaller cluster had a CV of 72.87%, which corresponded to a drastic decrease of inter-patient variability as the population mean CV was equal to 112.10%. All other clusters for each drug had only a single patient and therefore the CV could not be assessed. Clustering was compared to covariates of patients, such as gender, age and gene polymorphism, to see if there was any correlation however none was found.

## Discussion

Precision and personalized medicine requires accurate technologies for drug administration and proper systems pharmacology approaches for individual patient multidimensional data

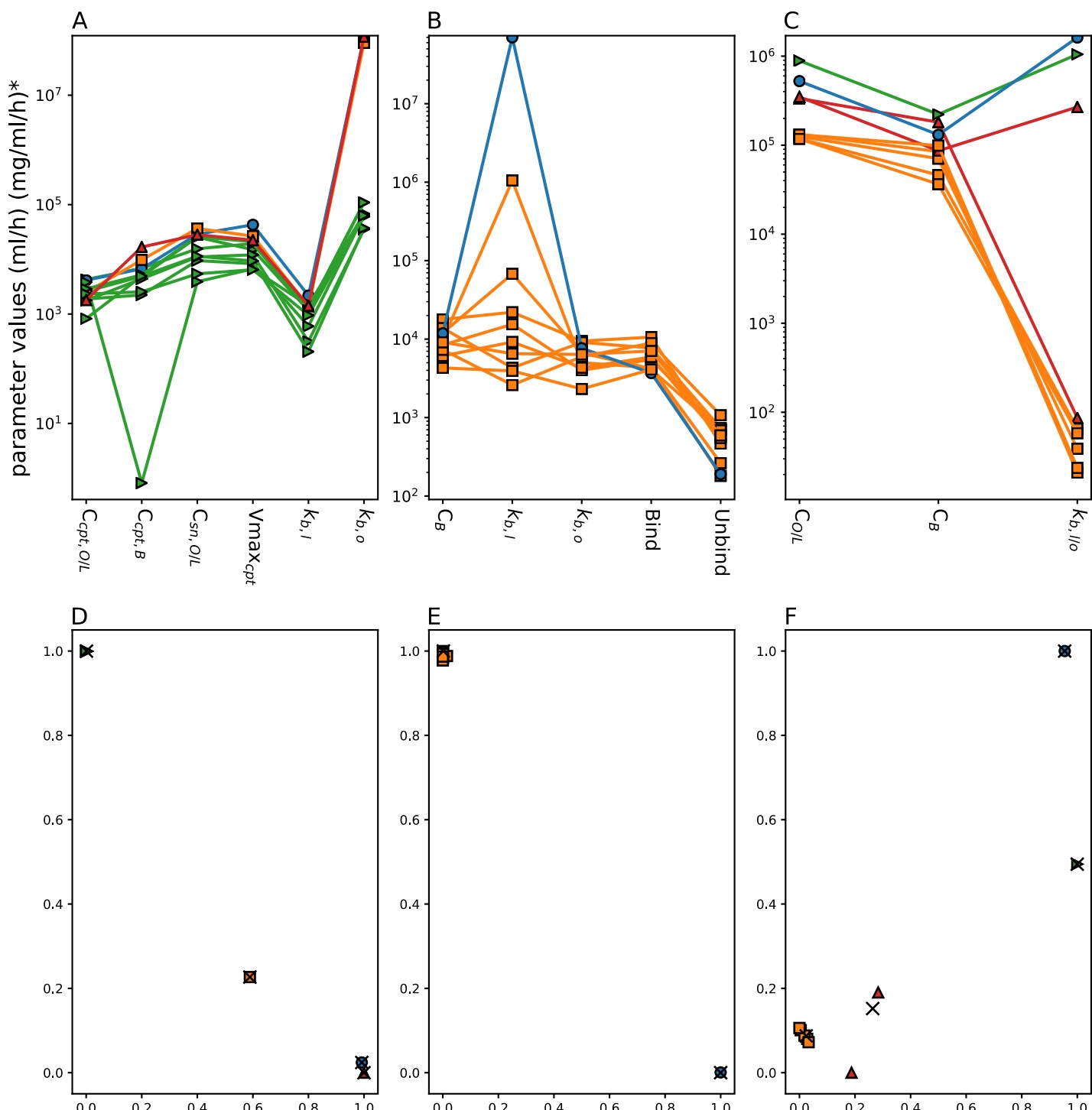

**Fig 9. Inter-patient variability in drug PK parameters.** The first line shows parameter variability across the considered patient population for irinotecan (A), oxaliplatin (B) and 5-fluorouracil (C), the colour and symbols represent the clusters each parameter set belongs to. The parameters are named with reference to the schematics of the models, the subscripts refer to the blood (B), organs (O) and liver (L). In the irinotecan parameters, additional subscripts cpt and sn refer to irinotecan and SN38 respectively. The second line shows multidimensional scaling representation of patient clustering based on their PK parameters for irinotecan (D), oxaliplatin (E) and 5-fluorouracil (F), the $x$ refer to the cluster centroids and the points refer to patient PK parameters projected onto 2D plot.

analysis. Here, plasma PK data of the OPTILIV trial in which patients received irinotecan, oxaliplatin and 5-fluorouracil through a chronomodulated schedule delivered by an infusion pump into the hepatic artery were mathematically analysed. To allow for an accurate analysis of PK patient data, a model of the pump drug delivery was successfully designed and connected to semi-mechanistic PK models. Although no data were available to directly validate the model-predicted drug infusion rates, the overall framework achieved a very good fit to individual time-concentration profiles which showed model accuracy. The validity of the approach was further demonstrated by the improved data fit using the PDE explicit solution connected to PK models compared to PK models directly integrating infusion profiles that were programmed into the pump. This study gave insights into inter-patient variability and paved the path to treatment optimization.

The simulations for the pump-to-patient model showed and quantified a delay between the actual start of the pump and the time when the drug appeared in the patient blood which was due to the delay needed for the drug solution to fill up the infusion tube and eventually reach the patient. A validation of this model prediction could be seen directly in the data as 5-fluorouracil and oxaliplatin plasma concentrations were close to zero for the first two times of measurements. The length of this delay depends on both the drug solution concentration and the volume of the infusion tube, so that its importance was high for oxaliplatin, intermediate for 5-fluorouracil and minor for irinotecan. Temporal accuracy is key for precision medicine especially in the context of chronotherapy and chronomodulated drug delivery. Thus, the programmation of any drug administration devices need to account for these delays. The pump-to-patient model that we present here allow to adapt any infusion schemes for any drug administration devices in order to properly administer the treatment schedules initially intended by the oncologists.

In addition to such "pump-to-body" delay, the increase in free Pt concentration near 22:00 shown in the PK data was explained by a spike in oxaliplatin delivery resulting from the glucose rinse flushing out the residual oxaliplatin left within the infusion tube. This phenomenon was well captured and quantified by oxaliplatin PK model which predicted that the quantity of drug delivered in the final spike was equal to 10.7% of the total dose. The model also showed that the $t_{max}$ of oxaliplatin plasma concentration was shifted by several hours due to this delivery profile spike. In silico simulations also predicted that the glucose flush would alter the PK of 5-fluorouracil. The spike only accounted for a small amount of 5-fluorouracil dose of 5.36% and may not have caused any significant detrimental effect. More data points covering the time of unexpected drug administration due to the glucose flush would have further validated the model which already achieved a very good fit to available data points. However, free oxaliplatin plasma concentration displayed complex patterns with high values at the start of the glucose flush for patient 1, 2, 3 and 7 which left no doubt on the large impact of the glucose flush on oxaliplatin administration. Similarly, unexpectedly high plasma concentrations of 5-fluorouracil were observed at the start of the glucose flush for patient 5 and 9 which partially validated the model. The delivery spike due to the glucose rinse did not seem to have influenced the plasma concentration profile of irinotecan because the drug concentration in the solution was much lower and the flow rate programmed into the pump was much higher as compared to oxaliplatin and 5-fluorouracil administration. Indeed, the spike only accounted for less than 2% of the total dose of irinotecan.

The pump-to-patient model further showed that these inconsistencies between the simulated and intended drug administration could be overcome with a simple and easily constructed adaptation of the infusion profiles, given the specific dimensions of the infusion tube. The new profile showed a much better match with the original intended administration profile.

Several published clinical studies propose mathematical models of the PK of 5-fluorouracil, oxaliplatin or irinotecan with various levels of complexity. First, a physiologically-based PK model of capecitabine, a pro-drug of 5-fluorouracil, was designed for humans [35]. However, the data available in the OPTILIV study would not allow for estimating parameters of such a detailed model. Next, numerous clinical studies have performed compartment analysis of plasma PK data from cancer patients receiving either 5-fluorouracil, oxaliplatin or irinotecan [41]. These models were designed for intravenous injection and could not be readily used for intra-arterial hepatic administration. Thus, the development of new semi-physiological PK models was necessary to include the drug delivery site as a separate compartment, that was different from the Blood compartment for which data was provided. Furthermore, the intention was also to develop more physiologically-relevant models in view of future account of circadian rhythms and chronotherapy optimization investigations. Indeed, the developed models are called semi-physiological as the compartment volumes together with relative fractions of clearance routes were inferred from literature. The quantity of data available for this study limited the models to being semi-physiological in nature. However, these models could be further extended to physiologically-based models, with increased data sets, by detailing the "Organ" compartment and being connected to mechanistic PD models to represent organ-specific drug PK-PD. Furthermore, the current models do not account for any circadian rhythms although they may largely impact on drug PK-PD. Thus, new circadian clinical studies are needed to improve the models towards drug chrono-administration optimization.

Inter-patient differences in maximum plasma drug concentrations and in the time at which it occurred led us to further investigate variability in between subjects. Irinotecan showed the lowest mean variability. Clustering analysis indicated that patients could be classified into five clusters with respect to irinotecan PK parameters. The second largest inter-patient variability was found for 5-fluorouracil. Clustering for 5-fluorouracil showed there was four clusters. Regarding oxaliplatin, there was the largest variability between patients PK model parameters with all parameters showing high variance. Clustering according to oxaliplatin PK parameters split patients into two clusters leading to isolate patient 7. This clustering of the patients led to a reduced inter-patient variability for all drugs, especially for irinotecan and 5-fluorouracil. This decrease in CVs is not unexpected, but the significant level of reduction means this method could be used as a way to stratify patients into treatment groups with less inter-patient variability in PK profiles. The measure of inter-patient variability could be interpreted as indicators of the need for personalisation as high differences between subjects implies high potential benefit of drug administration personalisation. Here, we demonstrated that the PK of all three considered drugs displayed important inter-subject variability. The remaining clinical challenge lays in determining clinical biomarkers for stratifying patients before drug administration, in order to reach the intended plasma PK levels. In order to do so, patient clusters were compared to known covariates such as age, gender and gene polymorphisms. However, none showed significant correlation. We then performed modelling analyses and identified the critical PK parameters for irinotecan, 5-fluorouracil and oxaliplatin which were the transport parameters between the Blood and either the Liver or the Organs compartments.

## Conclusion

In conclusion, a mathematical framework was designed to allow for accurate analysis of patient PK data. A model of the dynamics of the drug solution from the pump to the patient's blood was designed, irrespective of the drug delivery device. It was used to represent the chronomodulated drug administration though the Mélodie infusion pump into the patient hepatic artery of irinotecan, oxaliplatin and 5-fluorouracil. The model revealed significant inconsistencies

between the drug profiles programmed into the pump which corresponded to the drug exposure intended by clinicians and the actual plasma PK levels. Importantly, it allowed for the design of innovative drug in-fusion profiles to be programmed into the pump to precisely achieve the desired drug delivery into the patient's blood. Next, the pump-to-patient model was connected to semi-physiological models of the PK of irinotecan, oxaliplatin and 5-fluorouracil. The overall framework achieved a very good fit to data and gave insights into inter-patient variability in the PK of each drug. Potential clinical biomarkers for treatment personalisation were suggested although further investigations in larger cohorts of patients are required. Overall, this complete framework informs on drug delivery dynamics and patient-specific PK of irinotecan, oxaliplatin and 5-fluorouracil towards precise and personalized administration of these drugs.

## Methods

### Ethics statement

The pharmacokinetics data used in this investigation came from Lévi et al pharmacokinetic investigation [20] and the comparison study companion study of the European OPTILIV trial (ClinicalTrials.gov study ID NCT00852228), which involved nine centres in four countries [42]. The data has been analysed anonymously.

### OPTILIV clinical datasets

The OPTILIV trial included 11 colorectal cancer patients with liver metastases (7 men and 4 women with median age of 60). The combination of irinotecan, oxaliplatin and 5-fluorouracil was delivered to patients by Hepatic Artery Infusion (HAI) using the Mélodie pump [20]. The patients received an intravenous administration of cetuximab 500 mg/m2 over 2 h 30 min on the morning of day 1 which was not modelled. From day 2, chronomodulated HAI of irinotecan (180 mg/m2), oxaliplatin (85 mg/m2) and 5-fluorouracil (2800 mg/m2) were given (Fig 10). Irinotecan was delivered as a 6-h sinusoidal infusion starting at 02:00, with a peak at 05:00 on day 2. Oxaliplatin was administered as an 11h 30min sinusoidal infusion beginning at 10:15 with a peak at 16:00 on days 2, 3 and 4. 5-fluorouracil was also delivered as an 11h 30min sinusoidal infusion beginning at 22:15 with peak delivery at 04:00 at night, on days 3, 4 and 5. The superiority of this drug scheduling compared to non-circadian based administration was demonstrated for intravenous administration within several international clinical trials [37]. Between each drug infusion, there was a glucose serum flush which cleared the tubing. This was a 30-min sinusoidal infusion beginning at 09:45, and then again at 21:45 i.e. at the end of each infusion (Fig 10).

Plasma pharmacokinetics (PK) data was gathered after the first dose of irinotecan, oxaliplatin and 5-fluorouracil and measured longitudinally for each individual patient. Plasma concentrations of irinotecan and its active metabolite SN38 were determined, using high performance liquid chromotagraphy (HPLC), at the start of infusion, then at 2, 3, 4, 6, 8 h 15 min and 31 h 45 min post HAI onset, for a total of seven time points, including baseline. Oxaliplatin concentrations were determined by measuring platinum plasma levels using spectrophotometry, for both unbound and total platinum levels. Oxaliplatin binds to proteins in the blood and the free Pt fraction is the biologically active one. Thus, oxaliplatin concentrations were determined at the start time of infusion, then at 3, 6, 9 h, 11 h 30 min and 17 h 15 min post HAI onset, for a total of six time points, including baseline. Plasma concentrations of 5-fluorouracil were determined using HPLC, at the start of infusion, then at approximately 3 h, 5 h 45 min, 9 h and 11 h 30 min post HAI, for a total of five time points, including baseline. The exact

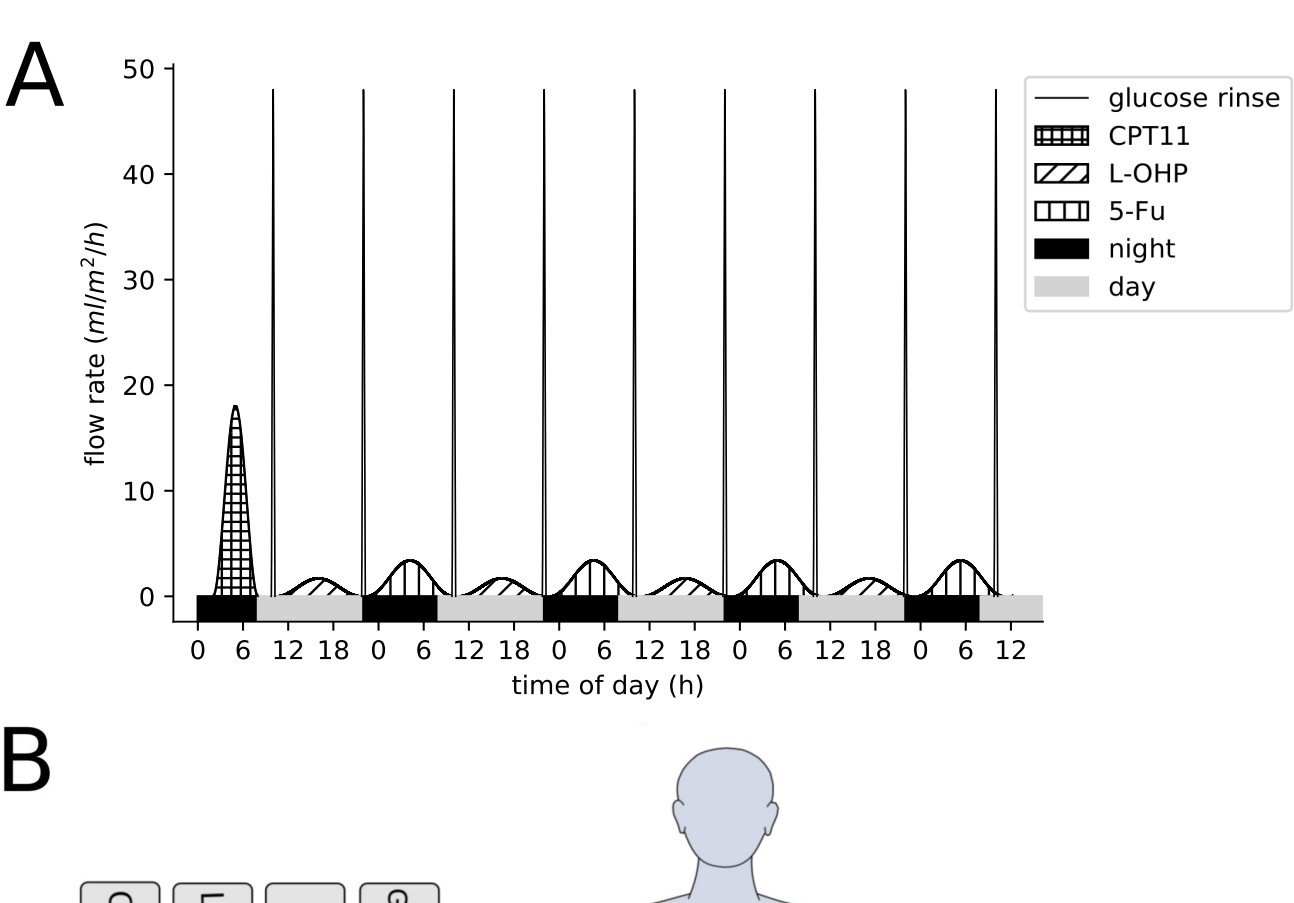

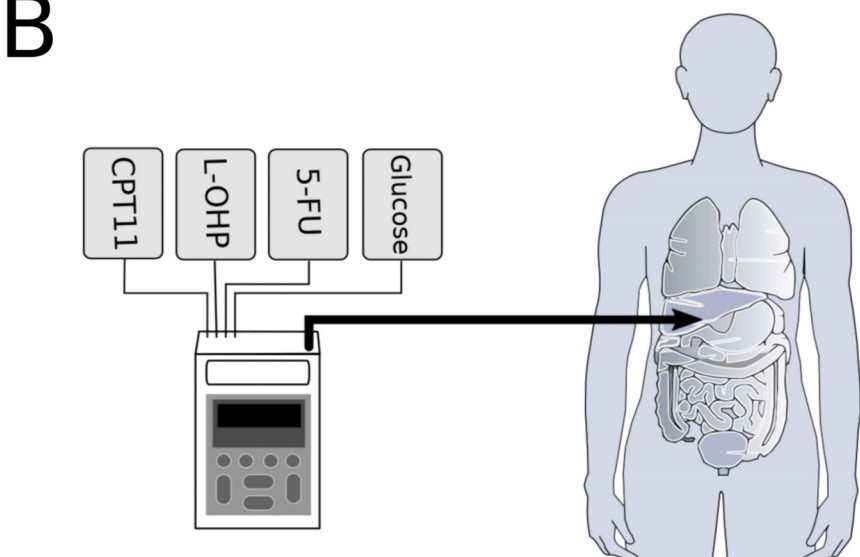

**Fig 10.** (A) Delivery profiles of irinotecan, oxaliplatin, 5-fluorouracil and glucose flushes as administered in the OPTILIV clinical trial. (B) Schematic of the Mélodie infusion pump (Axoncable, Montmirail, France) used in the OPTILIV study for hepatic artery infusion [20].

method used to assess plasma concentrations can be seen in Levi et al paper of the OPTILIV study [20].

## Pump description

The Mélodie pump system weighs 500 g when empty (excluding drug reservoirs and batteries) and measures 160 x 98 x 34 mm. The pump consists of four channels which correspond to the four reservoirs that are connected to the pump. Each reservoir can have a maximum volume of

2 L. The four channels are controlled by four independent mechanisms which control the delivery to the infusion tube (Fig 1). For the OPTILIV study, the infusion tube comprised of two sections, the first was 135mm long with a diameter of 2.5mm, and the second section was 1500mm long with a diameter of 1mm. The two sections had a total volume of 1.84ml. The four pump reservoirs were loaded with irinotecan, oxaliplatin, 5-fluorouracil and 5% glucose solution respectively, with the latter one being used for washes in between drug infusions [43].

## Mathematical modelling

A pump-to-patient mathematical model was designed as follows, irrespective of the drug delivery device. The drug solutions dynamics from the pump to the patient's blood was modelled using a Partial Differential Equation (PDE) considering time and 1 spatial dimension. This method was chosen as PDEs can take into account both time and space which was key for modelling systems such as pump delivery. The PDE was solved using a backward finite difference method written by the authors within Python 3.5.2 (https://www.python.org/). The drug PK models were based on Ordinary Differential Equations (ODEs) programmed using Python 3.5.2 and solved using the odeint function from the scipy library version '1.1.0' [44].

PK model parameter estimation involved a weighted least square approach, with conditions also placed on the drug clearance routes. For the fit of the data of a given patient, the residuals were weighted by a estimated measurement error of 10% inline with precision values of the assay methods [45–47]. This method allowed to correct the residuals to be of the same order of magnitude for the parent drug CPT11 and the metabolite SN38, or for oxaliplatin free and bound concentrations. The minimization of the least square cost function was performed by the Covariance Matrix adaptation Evolution Strategy (CMAES) within Python which has been shown to be successful at handling complex cost function landscapes [48]. Model goodness of fit was assessed using the sum of squared residuals (SSR) and $R^2$ values. PK model parameter numerical identifiability given the available data was investigated in a two-step process as follows. First, parameter sensitivity regarding the least-square cost function was computed via a global Sobol sensitivity analysis as a necessary condition for identifiability [49]. This method assesses the relative contributions of each parameter to the variance in the cost function obtained when parameter values are varied, and thus allows for the identification of parameters which have no effect on the cost function and are therefore not identifiable from the available dataset. This step allowed a first reduction of the PK models. Next, likelihood profiles of parameters of the reduced models were derived following the procedure outlined in [50]. Additional biological constraints derived from literature were added to ensure numerical identifiability of all parameters. This two-step model design process was undertaken as computing likelihood profiles is associated with a high computational cost.

PK models were fit to pooled data first to get an indication of general model fit then to single-patient plasma PK datasets independently to obtain patient-specific parameter values. Data was available for 10 to 11 patients which was too few to undertake mixed-effect population analysis and to reliably estimate the parameters variance within a patient population [51, 52]. Sampling points at 6 hours post injection for irinotecan and 11 hours 30 mins post injection for oxaliplatin and 5-fluorouracil theoretically occurred at the same time as the start of the 30 min glucose flush, that is 9:45 for irinotecan and 5-fluorouracil, and 21:45 for oxaliplatin. As described in the results section, the flush was equivalent to the administration of the drug quantity remaining within the tube and logically influenced plasma drug concentrations. However, the exact time of patient blood collection was not reported and could vary by 10 to 15 minutes due to clinical constraints. Hence, the information of whether the blood sample was taken before or during the flush was not available. Thus, the collection time of the data

points at theoretically 21:45 for oxaliplatin, 9:45 for irinotecan and 9:45 for 5-fluorouracil were unchanged if the drug concentration at the preceding data point was greater than the current one, indicating the flush might not have occurred yet. If not, the collection time was modified and set equal to the glucose peak time, which is 15min after its start time i.e. 22:00 for oxaliplatin and 10:00 for irinotecan and 5-fluorouracil, such value leading to the best model fit. Overall, the collection time was changed compared to the theoretical one for patients 1, 2, 3 and 7 for oxaliplatin, for patient 5 for 5-fluorouracil, and for no patients for irinotecan.

## Inter-patient variability and patient clustering based on PK parameters

Given the relatively small number of patients, the inter-patient variability in parameter values was assessed using a nearly unbiased estimator of coefficient of variation (CV),

$$CV = \left(1 + \frac{1}{4n}\right) \times \frac{\sigma}{\mu} \times 100,$$

where $\mu$ is the parameter mean, $\sigma$ the parameter sample standard deviation and n is the number of patients.

Next, fuzzy c-means clustering was used to define patient clusters based on individual PK parameters, for each drug separately. The fuzzy c-means clustering was done using a python library sckit-fuzzy version '0.2' (http://pythonhosted.org/scikit-fuzzy/). The method is based on the determination of cluster centroids and classification of patient parameter vectors into the clusters such that the following quantity is minimised:

$$\sum_{i=1}^{n}\sum_{j=1}^{c} w_{ij}^2 (x_i - c_j)^2$$

where $n$ is the number of patients, $c$ is the number of clusters, $x_i$ is the parameter vector of patient $i$, $c_j$ is the centroid of cluster $j$. $w_{ij}$ is the probability of patient $i$ belonging to cluster $j$ and can be expressed as:

$$w_{ij} = \frac{1}{\sum_{k=1}^{c} \left(\frac{x_i - c_j}{x_i - c_k}\right)^2}$$

Note that, for a given patient $i$, the following holds:

$$\sum_{j=1}^{c} w_{ij} = 1.$$

The validity function proposed by Fukuyama and Sugeno was used to determine the number of clusters for each drug. The function is defined as:

$$\sum_{i=1}^{n}\sum_{j=1}^{c} w_{ij}^2 (||x_j - c_j||^2 - ||c_j - \bar{c}||^2)$$

where $\bar{c}$ is the average of the centroids. The number of clusters were chosen between 2 and n-1 inclusively such that the $V_{FS}$ was minimised. Plotting the clustering results was done using a multidimensional scaling (MDS) algorithm which projects multidimensional data onto a 2D plane while keeping distance metric scaled relatively to original data (Python library sklearn. manifold [53]). Correlation coefficients between original Euclidean distance and 2D-Euclidean distance were calculated were high for all models ($> 0.98$) which showed that the MDS projections were accurate [54].

## Supporting information

**S1 Text. S1 contains model equations and parameters, identifiability information and plots, and clustering analysis details.**
(PDF)

## Acknowledgments

We would like to thank Sami Al-Izzi of the University of Warwick Mathematics Institute and Dr Thomas Lepoutre (Inria, team Dracula, Lyon, France) for their discussions.

## Author Contributions

**Conceptualization:** Francis Lévi, Annabelle Ballesta.

**Data curation:** Pasquale F. Innominato, Francis Lévi.

**Formal analysis:** Roger J. W. Hill.

**Methodology:** Roger J. W. Hill.

**Supervision:** Francis Lévi, Annabelle Ballesta.

**Writing – original draft:** Roger J. W. Hill, Pasquale F. Innominato, Francis Lévi, Annabelle Ballesta.

**Writing – review & editing:** Roger J. W. Hill, Annabelle Ballesta.

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
