## [Decision Letter · Decision Letter 0]

3 Sep 2019

Dear Dr Hill,

Thank you very much for submitting your manuscript 'Optimizing circadian drug infusion schedules towards personalized cancer chronotherapy' for review by PLOS Computational Biology. Your manuscript has been fully evaluated by the PLOS Computational Biology editorial team and in this case also by independent peer reviewers. The reviewers appreciated the attention to an important problem, but raised some substantial concerns about the manuscript as it currently stands. In fact, a reviewer recommended rejection. While your manuscript cannot be accepted in its present form, we are willing to consider a revised version in which the issues raised by the reviewers have been adequately addressed. We cannot, of course, promise publication at that time.

Sincerely,

James Gallo

Associate Editor

PLOS Computational Biology

Douglas Lauffenburger

Deputy Editor

PLOS Computational Biology

[LINK]

Reviewer's Responses to Questions

**Comments to the Authors:**

Reviewer #1: Precision medicine not only requires personalized drugs and dosage according to the patient's genotype in the upstream but also needs accurate technologies for drug administration in the upstream. Instead of focusing on the upstream analyses as many other researchers do currently, the authors of this article illustrate how to implement a systems pharmacology approach for downstream analysis of precision medicine: optimizing drug infusion pumps.

Due to the large time delays between the actual pump start and the time of drug detection in patient blood, the authors first built a pump-to-patient mathematical model, confirmed it by pharmacokinetics data, and then clustering patients according to the parameters of each patient's prediction model. These research processes are desirable; however, the quality of this article and other issues make it not be satisfied with the criteria of PLoS Computational Biology.

Major issues:

1. Wrong numbers. For examples, the mean Vmax of the irinotecan PK model (SI Table 2) should be about 2.41e05 instead of 5.96e04, and thus CV is not correct, which also affect the results shown in the main article; the mean "Cl, Co" of the oxaliplatin PK model (SI Table 5) should be 687.9 instead of 68.80; the mean Cb of the oxaliplatin PK model (SI Table 5) should be 544.2 instead of 54.42.

2. Not appropriate statistics. For examples, when calculating CV, the authors used population standard deviation instead of sample standard deviation; using average improvement of separated SSR, CV, and R-square might not be the best choice, and the authors could try pooling the patients' data together and recompute the overall improvement, CV, and R-square.

3. Overestimated prediction performance. In Figure 6-8 (actually they are Fig 7-9 shown in my downloaded PDF file) and SI Fig 1-3, only a few data points cover the tail region. Since this article emphasize the delay effect of drug delivery in the traditional approach, more data should be collected during this period, if possible.

4. Not readable figure index. Figure 1-9 mentioned in the article are Figure 2-10 in my PDF file; Figure 10 mentioned in the article is Figure 1 my PDF file. I think the "Fig 1" in line 393 is Figure 10 mentioned in the article and is Figure 1 my PDF file. If this PDF version is different from what the authors can download from the system after they submitted, please ignore this comment.

5. Inconsistent parameters. Comparing the formula of Fukuyama and Sugeno (VFS) shown in the reference [31] and that shown in this article (at the bottom of P.14), the authors chose 1 for the parameter m, the sensitivity to the fuzzifier; however, the formula shown in the middle of P.14 assume the m = 2. This is either a typo or should be explained.

6. In Figure 1 (Fig 2 in my PDF file), the curves of intended profile and simulated profile in sub-figure C, E, G are not shown correctly and these two curves should be swapped.

7. In line 248, it says "Blood-Liver and Blood-Organs transport parameters presented the highest CVs for all three drugs", but in SI Table 2, the CV of "Efl, Upl" is 53.97, which is not the highest two; in SI Table 5, the CV of "Efo, Upo" is 41.15, which is not the highest two.

8. Figure 3-5 (Fig 4-6 in my PDF file) show three independent models, but Figure 6-8 (Fig 7-9 in my PDF file) show the concentrations of the three anticancer drugs are not 0 at many time points. Do they have interaction, either physically or statistically? Should the three models be merged?

Minor issues:

1. The drug rate curve of the new delivery method shown in Figure 2 (Fig 3 in my PDF file) is still a simulation result, which has not been validated in this article.

2. Which Python package and version was used for solving PDE (line 431)? What's the version of scipy (line 435)? What's the version of sckit-fuzzy (between line 477-478)?

3. Typo: "vhronotherapy" in line 335 should be "chronotherapy"; "sum of square residuals" in the main article and supplementary should be "sum of squared residuals"; "effux" in Fig 3-5 (Fig 4-6 in my PDF file) should be "efflux".

4. Format issues: "10.7\\%" -> 10.7%; "tmax" in line 307; "VFS" in line 479 (FS should be subscript); "(> 0.98)" in line 484 cannot be shown correctly.

5. The hyperlink shown in line 53 should be converted into an official reference format.

6. Inconsistent sub-figure index. In the article, they are in lower case; in the figures, they are upper case.

7. The link in reference [20] doesn't work. It should be "" ext-link-type="uri" xlink:type="simple">https://books.google.co.uk/books?id=5Pv4LVB_m8AC"

Reviewer #2: Title: Optimizing circadian drug infusion schedules towards personalized cancer chronotherapy.

Authors: Roger J. W. Hill et al.

Brief summary:

This manuscript describes the results of a modeling exercise using pharmacokinetic (PK) data obtained in a clinical study investigating the “personalized chronotherapy” of intrahepatic infusions of three anticancer drugs, i.e., irinotecan (CPT-11), oxaliplatin (OXI) and 5-fluoro-uracil (5-FU), in patients with colorectal cancer and liver metastases. The three drugs of interest were infused at variable, time-dependent rates using a novel programmable infusion pump and were timed chronopharmacologically, i.e., at different peak times. During the first treatment course, a total of 6-8 venous blood samples were drawn during and after the individual respective infusions, and plasma concentrations of CP-11 (prodrug), SN-38 (active metabolite of CP-11), bound and unbound OXI and 5-FU were measured used validated LC methods (as described in the original OPTILIV reference).

In analyzing the PK profiles in 11 patients, he authors proceeded to develop two mathematical models: (a) a “pump-to-patient” model, modeling the delay in achieving intrahepatic concentrations relative to the intended/nominal infusion rate profile due to the dead space of the infusion tubing, given the slow (6-8 hours) infusion rates using a time- and space-dependent differential equation, with tube length, fluid velocity/volume flow rate and drug concentrations fixed for each drug of interest, and (b) a set of “semi-mechanistic” PK models for each of the five plasma analytes of interest, i.e., for the most part 3-compartment models (liver, blood, “organs”) connected by first-order processes (except the formation of SN-38 from CPT-11, modeled as Michaelis-Menten reaction) and irreversible, first-order elimination occurring from 2-3 compartments. Compartments and processes were interpreted mechanistically, based on available in-vitro and in-vivo literature information on the ADME for each analyte of interest. The final models were implemented as sets of differential equations for each compartment for each analyte. Statistical criteria for parameter identifiability and mechanistic plausibility were used to reduce the number of parameters and restrict their value space, obviating the need to estimate them from the limited data. Weighted nonlinear regression was used to estimate the final parameters from each individual patient profile for each of the three drugs of interest.

Using the “pump-to-patient” model for each of the three drugs, the authors were able to simulate infusion rates/concentration profiles reaching the liver compartment, demonstrating the lag-time relative to the nominal/programmed infusion rates, not only during the target infusion times, but also during the terminal ”flush” (namely clinicians attempting to clear the infusion tubing of the drug, which contributed up to 11% of the total dose for OXI); this delay was most significant for OXI - for reasons not further elaborated on in the manuscript. Furthermore, using their “semi-mechanistic” PK models (see below), the authors were able to show successfully that using the simulated rather than the nominal infusion rate profiles improved the goodness of fit, supporting the importance and validity of the “pump to patient” model.

Using their various final “semi-mechanistic” PK models with a restricted number of presumably estimable parameters (3-6), the authors were able to obtain acceptable/good fits of the available plasma concentration-time profiles for the five analytes. Population variability in parameter estimates was very large, and no apparent attempt was made to interpret the obtained parameter estimates (primarily elimination and inter-compartmental transfer clearances) in mechanistic/physiological terms or identify any patient characteristic(s) as a source of the population variability.

Overall, the authors concluded that the “pump-to-patient” model and “semi-mechanistic PK model provide a framework to optimize drug infusion through hepatic artery delivery and to analyze inter-patient variability in drug PK.

Overall assessment and Recommendation

The authors have adequately addressed one important topic of interest for a better understanding of the inter-patient variability in the PK of CPT-11, OXI and 5-FU after slow, time-dependent infusion rates by developing and validating their “pump-to-patient” model, using a mathematical model that allows the conversion of ANY nominal (“pump”) infusion rate scheme into the actual (“patient”) infusion rate, considering drug concentration, volume flow rate and tube volume. This approach can be generalized beyond the particular study for any infusion regimen, not only via hepatic artery.

However, their “semi-mechanistic” PK modeling attempts were not able to overcome the major limitations of the available PK data sets, namely only 4-7 quantifiable plasma concentrations, sampled primarily during the infusion with only 1-2 samples after the end of the infusion (i.e., able to provide information about systemic drug disposition, uncontaminated by drug administration). In addition, despite their attempts to reduce the number of model parameters, the final PK models still estimated ~1 parameter for each data point, resulting in over-parameterization of the models. Finally, the authors appear to have misinterpreted some of the available ADME information as they incorporated it in their modeling, building assumption upon assumption in their final models; in particular, the authors seem to be unaware that in addition to parameter identifiability, model identifiability is a major issue for linear compartmental models in assigning elimination pathway(s) to central/peripheral body compartments. Their travails are further complicated by nontraditional, inconsistent nomenclature in naming model compartments and parameters, apparently incorrect/incomplete differential equations and inappropriate units for their parameter estimates as well as sloppy numbering of their figures and discrepancies between text, figures, tables, model schemes and equations - which made it extremely difficult and time-consuming for this reviewer in trying to fully understand their methods and results and arrive at a proper assessment of the merits of the manuscript and make the recommendation below!

Therefore, the overall recommendation by this reviewer is to reject the manuscript from publication. For the benefit of the authors with future semi-mechanistic PK modeling, extensive individual concerns/recommendations are explained and summarized below.

Major Concerns:

Text:

1. Most plasma samples (4-7 time points) were collected during infusion, very few after the end of the infusion. Two post-infusion samples for CPT-11, only one for OXI and zero for 5-FU, which makes it difficult for the parameter estimates of a complex 3-6 compartmental modeling, especially the systemic drug disposition processes. For example, in the OPTILIV study (reference no.19), the plasma levels of OXI for most patients did not show any declining trend, which makes it difficult to estimate clearance-related parameters.

2. “Semi-mechanistic” PK modeling:

The following issues applied to the final PK models of all three drugs:

a) Equations (SI 1.1, 1.2 and 1.3) do not match their semi-physiological models (Figure 3-5); for 1.1. and 1.2. first DE, it appears that the opening parenthesis is missing, as ALL terms should be divided by V.

b) All units for the parameter estimates listed in SI (Table 1,2, 5,8) are incorrect: For example, Ci is the equivalent of clearance from compartment I, relating change in mass/time to concentrations; therefore, the unit cannot be mg/h and it does not match Figure 10 in the main paper body either.

c) The number of parameters they estimate listed in the SI does not match their equations and models.

d) Line 159-161: They “individualized” volume in all compartment for each patient but did not explain how they did that and no fixed or estimated values reported. In the main text, “scaling” of physiological volumes is mentioned, but not elaborated further. Regardless, those were model parameters that were fixed arbitrarily, especially for the “Organ” compartment.

e) Line 165: The compartmental physiological model includes three compartments, Blood, Organ and Liver. Elimination clearances include renal elimination for the Blood compartment, intestinal elimination for the Organs compartment and biliary excretion for the Liver compartment; apparently no hepatic metabolism (other than SN-38 formation) is allowed. However, biliary excretion does not occur for each drug. The authors should list explicitly what the contribution of biliary excretion for each of the three drugs is assumed to be to total clearance from in-vivo ADME information. In addition, other than biliary excretion, how about liver metabolism, for many drugs, liver metabolism is non-negligible clearance. For example, CPT-11 mainly metabolized by carboxylesterase and CYP450 (ref..25, Figure 1).

f) Line 165: The authors didn’t clearly explain how they differentiate elimination clearance by intestinal secretion from hepatobiliary excretion. Since the 3 drugs were administered directly into the hepatic artery, this administration route excludes the possibility of the drug metabolism in the gut before entering in the system. The only possible reason of intestinal elimination would be biliary excretion of the parent drug, and then the drug may undergo further intestinal metabolism in the gut. However, this type of intestinal elimination would be counted as biliary excretion rather than intestinal elimination. It is not clear what the intestinal elimination in the physiological model is supposed to represent.

g) Line 171: “For the sake of simplicity, uptake and efflux rate constants were assumed to be equal for Blood-liver and Blood-Organs transport”. Any evidence or in-vivo/in-vitro data to make such assumptions? No relevant references are cited in the paper. More general, the terms “uptake” and “efflux” typically assume drug transport across biological membranes, while the in-vivo transfer across compartments also involved blood flows/organ perfusion; a more general terminology should have been used.

h) Line 175: Why was organ clearance assumed to be equal to liver clearance? Ref. 24 was not properly cited here. No relevant references cited in the paper support the assumptions of organ clearances equal to liver clearances for CPT-11 and OXI.

1) CPT-11

(a) For the transfer/conversion between the CPT-11 and SN-38 compartments, the authors used mass concentrations rather than molar concentrations, assuming the same molecular weight for CPT-11 and SN-38. This is incorrect!

(b) The authors assumed that volumes of CPT-11 and SN-38 are the same (see equation SI 1.1 (1) and (2), applying to Blood compartment and Organ compartment, too). No evidence is provided to support these assumptions.

(c) CPT-11 is the prodrug for SN-38, converted by carboxylesterase (CE) in the liver. Other than Liver, the authors also assumed CPT-11 is metabolized in Blood and Organ, and they share the same Vmax and Kcp values. No literature is provided to support this assumption

(d) Line 197-198: the authors took the Km value (presumably Kcp used in equations, authors need to make their parameters consistent) from an in-vitro study using Caco-2 cells, not hepatocytes. On top of that, no text indicates that if and how they did in-vitro-in-vivo extrapolation from cited in-vitro study. Additionally, the plasma/blood concentrations (and possibly other compartments as well) are below the Kcp value, suggesting that the MM reaction may be reduced to a simple first-order reaction (consistent with the observed PK dose-proportionality for CPT-11 after IV clinical doses, see refs 25 and 26). Finally, by fixing vmax, Kcp (or more likely vmax/Kcp), the authors are likely ignoring a major source of intra-/interpatient PK variability in CPT-11 PK - as they cannot estimate it for each patient separately!

(e) Line 186: “34% renal elimination pathway” supported by a mass balance study with I.V. infusion of 125 mg/m2 CPT-11 to cancer patients (ref. 25). However, 51% of the total administered dose excreted by bile is not a proper statement. From their cited reference, “Fecal excretion accounted for 63.7±6.8 (range: 54.2–74.9%, n=7) of dose, whereas urinary excretion accounted for 32.1±6.9% (range: 21.7–43.8%; n=7) of dose.” Fecal excretion does not mean 100% parent drug excreted through bile. CPT-11 is mainly metabolized by the liver to SN-38 and also by CYP450 (supported by ref. 25, Figure 1).

(f) Line 188: The authors set the parameters to 15% of the total CPT-11 dose eliminated either through liver or organ and got the same final liver and organ clearance for SN-38 (SI Table 1.), which is a not physiologically relevant estimate (ref. 25).

(g) SSR for CPT-11 PK model (SI, Page 3). Presumably CPT-11 and SN-38 plasma concentrations were fit simultaneously to obtain the fits shown in Fig 1. And parameter estimates in Tables 1. and 2. In scaling SSR, how did the authors incorporate the large different concentration scale between CPT-11 and SN-38?

2) OXI:

(a) The semi-physiological model the authors used assumed first-order association and dissociation, i.e., reversible binding, of free and bound OXI, characterized by parameters “b” and “u”. However, literature reports that platinum binds irreversibly to plasma proteins and covalent binding to tissue (ref. 26). The authors seem to misinterpret the cited literature and did not incorporate the irreversible binding in their model.

(b) Line 190-191: “The amount of Pt bound within the Organs or within the Liver was set to 84% and 12% of the total dose, respectively.” The authors set the parameters of 84% and 12% from a study in mice (ref. 27) without discussing potential inter-species differences. Regardless, it is not clear how the authors came up with 84% and 12% from the cited paper.

(c) Clinical data show OXI to be equally cleared by tissue distribution and glomerular filtration from plasma (ref. 27). However, the authors assumed equally “efflux” and “uptake” to Organ. Their parameter estimates of blood clearance (Cb) and organ uptake (Upo) in SI table 5 showed very different values, which contradicts the results from the clinical study. The authors assumed equal liver and organ clearance without citing any clinical evidence in support.

(d) Line 205: the authors assumed the same PK properties between metabolite and OXI? Is there any evidence or relevant references to support this assumption?

3) 5-FU

(a) Line 193: The authors assumed 80% of total dose cleared through (“though” in text need to be corrected to “through”) liver. According to their cited ref. 24, “5-FU is eliminated primarily by hepatic metabolism, with less than 5% of the drug excreted unchanged in the urine”; the statement from the manuscript that “5-FU was shown to be mainly cleared through hepatic metabolism” is true. However, considering their early text (Line 165), they set liver biliary excretion to 80% for 5-FU is not proper. On top of that, they assume the same liver and organ clearance (see SI table 8), which does not match clinical findings.

Figures:

1. Figure 4: PK model of CPT-11: Blood compartment for SN-38 shows no clearance from blood (no clearance arrow CSN,B from BloodSN38), but the equation in SI (Eq1_(4)) does indicate a clearance of SN-38 from blood compartment (…-Csnb *Bsn). SAs in the text Line 200, SN-38 is eliminated by UDP-glycosyltransferases. These discrepancies need to be corrected.

2. Figure 5: PK model of OXI: There is an efflux arrow from Organ to Blood compartment for free OXI. However, no corresponding efflux from Organ to Blood for free OXI is shown in Equation 1.2 (9) in their SI. The authors either need to correct the model scheme or equation to make sure it is consistent.

3. Figure 6: There is no organ clearance in the scheme of their 5-FU model, which is consistent with what they explained in the text (Line 193) that 5-FU is mainly eliminated by liver through metabolism. However, the equation in SI 1.3 (15) shows organ clearance, so as their parameter estimate table SI table 8. This discrepancy needs to be corrected

4. SI Figure 5 (b) and (d) show large range for that parameter estimate. How did the authors justify values in those cases?

Minor Concerns/Recommendations:

Authors need to make sure that the symbols/abbreviations of their model parameters are consistent across text, equations, schemes, and tables!

Text

1. Line 124. The time delay for 5-FU is 2 h 202 min. Is it a typo of 202 min?

2. Line 186. It is the first time that SN-38 was written in the manuscript, however, the explanation of SN-38 shows later at Line 195.

3. Line 190. It is the first time that Pt was written in the manuscript, however, the explanation of Pt shows later at Line 202.

4. Line 164: Linear kinetics? Any references to support these assumptions?

Irinotecan follows linear PK, (conformed from the FDA drug label “Over the recommended dose range of 50 to 350 mg/m2, the AUC of irinotecan increases linearly with dose”) The linear assumption for irinotecan is true, but authors need to either justify in text or cite references properly. Apply to the other two drugs too, their linearity needed to be justified or properly cited.

5. Are the doses they used clinically relevant?

Yes, they are clinically relevant dosing regimen (confirmed by FDA drug labels of irinotecan and oxaliplatin and the reference of “Metzger G, et al. Spontaneous or imposed circadian changes in plasma concentrations of 5-fluorouracil co-administered with folinic acid and oxaliplatin: relationship with mucosal toxicity in patients with cancer. Clin Pharmacol Ther. 1994;56(2):190–201.”) However, authors need to properly justify the rationale of the doses they’ve chosen and cite the primary references..

6. Line 403-414: Analytical method and assay validation were not explained or properly cited from the OPTILIV Study.

7. Line 198: Although they use MM PK, but the highest detected SN38 concentration in the OPTILIV paper are much smaller than its km (supported by reference No.28). Therefore, it is an apparent first order transfer from CPT-11 to SN38. It is okay to still use MM equation, but instead if they use linear PK, it could give them a simpler model and an acceptable way to reduce their over-parameterization problem.

Figures

1. The numbers of all figures were not correctly cited in the text. For example, page 5, Figure 1 (a) shows OXI concentration profile in the infusion tube. However, it should be Figure 2. This also applies to other figures.

2. Figure 2 (supposed to be Figure 3 in text): What about the (volume) flow rate? With only information of amount rate, there is no way to differentiate flow rate and drug dose, please also include flow rate figures. More general, there should be table for each of the three drugs with: total dose, drug concentration in infusion solution, total volume infused, total infusion time as well as peak and average flow [mL/min] and dose rate [mg/hr]. Considering the tubing dead space (1.85 mL, as mentioned on page 5), this information may better put in perspective/explain, why the delay is worst for OXI and least for CPT-11 - as shown on this and the subsequent figure.

3. Figure 2 C, E G, the intended/nominal and simulated profiles have been mislabeled. Otherwise, it is not indicating lag-time, but action before drug on board. If so, what does that mean then?

Equations

1. What does “d(t)” represent? Presumably the input function in amounts/time?

2. Equations 1 and 7 are incomplete with missing open parentheses.

Reviewer #3: In this manuscript, Hill et al develop a model for chronotherapeutic infusion of cancer chemotherapy agents through a pump for optimized sinusoidal delivery. The authors first describe a model for delivery by pump to eliminate flow and volume dependent mismatch between the desired drug delivery and actual delivery to the bloodstream. Subsequently, pharmacokinetic modeling was performed using a three-organ model (blood, liver, and other organs) resulting on either six compartments or three depending on the compound. Finally, heterogeneity amongst the patient population was determined, and shown that only 3 of the patients were consistently clustered, highlighting the importance of individualized profiling.

Overall the manuscript is reasonably well written and clear, although there are certainly changes that could be made to improve readability and provide the reader with stronger intuition with regards to the importance of the findings.

Specific comments:

1. It might be useful to describe in the introduction why data from the OPTILIV study was chosen as opposed to other studies which are better parameterized. Does this study reflect commonly available information which might lead to better generalizability of the approach?

2. The introduction describes significant sexual dimorphism with regards to specificity in previous trials – is this something that might be addressed in the PK modeling? Was gender a factor in the clustering of inter-individual PK differences? It would seem that this is an important factor that must be considered in some way, and merits some discussion if not explicitly examined in this study.

3. In the introduction, the authors state that “Chronotherapy requires the error in drug infusion timing not to be greater than few minutes” – what is the evidence for this? Metabolic studies suggest that oscillations are the order of hours, so how and why must this so called chronotherapeutic window be narrowed so strictly?

4. Does the use of glucose in the final flush have an impact on any of the key parameters which have been determined as glucose will have significant impacts on metabolism.

5. With regards to the PK modeling, it is noted that the effect of circadian rhythms was neglected to avoid over-parameterization. While this is reasonable, it is also somewhat inconsistent with the earlier statement on the top of page 3 that ADME profiles can have several folds difference within a 24 cycle. Thus the limitations of not including the circadian profile must be clearly stated.

Minor comments:

‘Figure 1’ on page 5 appears to correspond to the actual Figure 2—not clear what the actual Fig 1 represents, particularly Fig 1A. In fact, all figures are misnumbered by 1, with the actual Fig 1 not referenced in the text.

Fonts on real Figure 4 are distorted (e.g. efflux reads as eff ux etc.).

X-axis on real Figure 10 is upside down

Line 335 vhronotherapy -- chronotherapy

**Have all data underlying the figures and results presented in the manuscript been provided?**

Reviewer #1: Yes

Reviewer #2: No:

Reviewer #3: Yes

PLOS authors have the option to publish the peer review history of their article (what does this mean?). If published, this will include your full peer review and any attached files.

Reviewer #1: No

Reviewer #2: Yes: Jurgen Venitz, MD, Ph.D., Professor, VCU School of Pharmacy

Reviewer #3: No

---

## [Decision Letter · Decision Letter 1]

22 Oct 2019

Dear Dr Hill,

Thank you very much for submitting your manuscript, 'Optimizing circadian drug infusion schedules towards personalized cancer chronotherapy', to PLOS Computational Biology. As with all papers submitted to the journal, yours was fully evaluated by the PLOS Computational Biology editorial team, and in this case, by independent peer reviewers. The reviewers appreciated the attention to an important topic but identified some aspects of the manuscript that should be improved.

We would therefore like to ask you to modify the manuscript according to the review recommendations before we can consider your manuscript for acceptance. Your revisions should address the specific points made by each reviewer and we encourage you to respond to particular issues Please note while forming your response, if your article is accepted, you may have the opportunity to make the peer review history publicly available. The record will include editor decision letters (with reviews) and your responses to reviewer comments. If eligible, we will contact you to opt in or out.raised.

- Supporting Information uploaded as separate files, titled 'Dataset', 'Figure', 'Table', 'Text', 'Protocol', 'Audio', or 'Video'.

We hope to receive your revised manuscript within the next 30 days. If you anticipate any delay in its return, we ask that you let us know the expected resubmission date by email at ploscompbiol@plos.org.

Sincerely,

James Gallo

Associate Editor

PLOS Computational Biology

Douglas Lauffenburger

Deputy Editor

PLOS Computational Biology

[LINK]

Reviewer's Responses to Questions

**Comments to the Authors:**

Reviewer #1: Most of the reviewers' concerns have been addressed.

Some minor issues/errors still need to be curated before being published:

1. The "wij" of the formula between line 575-576 should be "wij^2". Since the formula between line 569-570 in this article is equivalent to the formula (7) in Ref [40], the m=2 is used in this article; the formula between line 573-574 in this article is equivalent to the formula (5) or (10) in Ref [40], which further shows that the wij in this article is actually the uij in Ref [40]. However, the formula (17) in Ref [40] shows the uij has an exponent m, but the formula between line 575-576 in this article shows the wij has no exponent m. If m=2 is used, "wij" should be "wij^2".

2. The description in line 311-314 is not correct. It should be "In all three models the parameters which showed the largest inter-patient variability were transport parameters specifically, for irinotecan Blood-Organ transport (CV = 176.3), for oxaliplatin Blood-Liver transport (CV = 318.2) and Blood-Liver/Organ transport for 5-fluorouracil (CV = 187.52)."

3. Figure 5 still shows "eff ux" instead of "efflux"

4. Ref [21] was not the only hyperlink which doesn't work. Ahthough it is fixed, the hyperlinks in Ref [39] and [44] still don't work.

Reviewer #3: The authors have addressed all concerns adequately.

**Have all data underlying the figures and results presented in the manuscript been provided?**

Reviewer #1: Yes

Reviewer #3: Yes

PLOS authors have the option to publish the peer review history of their article (what does this mean?). If published, this will include your full peer review and any attached files.

Reviewer #1: No

Reviewer #3: No

---

## [Decision Letter · Decision Letter 2]

21 Nov 2019

Dear Dr Hill,

We are pleased to inform you that your manuscript 'Optimizing circadian drug infusion schedules towards personalized cancer chronotherapy' has been provisionally accepted for publication in PLOS Computational Biology.

In the meantime, please log into Editorial Manager at https://www.editorialmanager.com/pcompbiol/, click the "Update My Information" link at the top of the page, and update your user information to ensure an efficient production and billing process.

One of the goals of PLOS is to make science accessible to educators and the public. PLOS staff issue occasional press releases and make early versions of PLOS Computational Biology articles available to science writers and journalists. PLOS staff also collaborate with Communication and Public Information Offices and would be happy to work with the relevant people at your institution or funding agency. If your institution or funding agency is interested in promoting your findings, please ask them to coordinate their releases with PLOS (contact ploscompbiol@plos.org).

Thank you again for supporting Open Access publishing. We look forward to publishing your paper in PLOS Computational Biology.

Sincerely,

James Gallo

Associate Editor

PLOS Computational Biology

Douglas Lauffenburger

Deputy Editor

PLOS Computational Biology

Reviewer's Responses to Questions

**Comments to the Authors:**

Reviewer #1: All the reviewers' concerns have been addressed.

Reviewer #3: No further comments

**Have all data underlying the figures and results presented in the manuscript been provided?**

Reviewer #1: Yes

Reviewer #3: None

PLOS authors have the option to publish the peer review history of their article (what does this mean?). If published, this will include your full peer review and any attached files.

Reviewer #1: Yes: Liang-Chin Huang

Reviewer #3: No

---

## [Editor Report · Acceptance letter]

15 Jan 2020

PCOMPBIOL-D-19-01041R2 

Optimizing circadian drug infusion schedules towards personalized cancer chronotherapy

Dear Dr Hill,

I am pleased to inform you that your manuscript has been formally accepted for publication in PLOS Computational Biology. Your manuscript is now with our production department and you will be notified of the publication date in due course.

With kind regards,

Laura Mallard
